# Local, calcium- and reward-based synaptic learning rule that enhances dendritic nonlinearities can solve the nonlinear feature binding problem

**Zahra Khodadadi[1]\*, Daniel Trpevski[1], Robert Lindroos[1†], Jeanette Hellgren Kotaleski[1,2†]**

[1]Science for Life Laboratory, Department of Computer Science, KTH Royal Institute of Technology, Stockholm, Sweden; [2]Department of Neuroscience, Karolinska Institutet, Stockholm, Sweden

**\*For correspondence:** zahra.khodadadi@scilifelab.se

[†]These authors contributed equally to this work

## eLife Assessment

This computational modeling study builds on multiple previous lines of experimental and theoretical research to investigate how a single neuron can solve a nonlinear pattern classification task. The revised manuscript presents **convincing** evidence that the location of synapses on dendritic branches, as well as synaptic plasticity of excitatory and inhibitory synapses, influences the ability of a neuron to discriminate combinations of sensory stimuli. The ideas in this work are very interesting, presenting an **important** direction in the computational neuroscience field about how to harness the computational power of "active dendrites" for solving learning tasks.

**Abstract** This study investigates the computational potential of single striatal projection neurons (SPNs), emphasizing dendritic nonlinearities and their crucial role in solving complex integration problems. Utilizing a biophysically detailed multicompartmental model of an SPN, we introduce a calcium-based, local synaptic learning rule dependent on dendritic plateau potentials. According to what is known about excitatory corticostriatal synapses, the learning rule is governed by local calcium dynamics from NMDA and L-type calcium channels and dopaminergic reward signals. In order to devise a self-adjusting learning rule, which ensures stability for individual synaptic weights, metaplasticity is also used. We demonstrate that this rule allows single neurons with sufficiently nonlinear dendrites to solve the nonlinear feature binding problem, a task traditionally attributed to neuronal networks. We also detail an inhibitory plasticity mechanism that contributes to dendritic compartmentalization, further enhancing computational efficiency in dendrites. This in silico study highlights the computational potential of single neurons, providing deeper insights into neuronal information processing and the mechanisms by which the brain executes complex computations.

## Introduction

Classically, single neurons in the nervous system have been thought to operate as simple linear integrators such that the nonlinearity of dendrites can be neglected (*McCulloch and Pitts, 1943*). Based on this simplification, powerful artificial neural systems have been created that outperform humans on multiple tasks (*Silver et al., 2018*). However, in recent decades, it has been shown that active dendritic properties shape neuronal output and that dendrites display nonlinear integration of input signals (*Antic et al., 2010*). These dendritic nonlinearities enable a neuron to perform sophisticated

computations (*Tran-Van-Minh et al., 2015*; *Gidon et al., 2020*), expanding its computational power beyond what is available with the somatic voltage threshold and making it similar to a multilayer artificial neural network (*Poirazi et al., 2003*).

A dendritic nonlinearity common among projection neurons in several brain areas is the NMDA-dependent plateau potential (*Oikonomou et al., 2014*). Plateau potentials are regenerative, all-or-none, supralinear voltage elevations triggered by spatiotemporally clustered glutamatergic input (*Schiller et al., 2000*; *Polsky et al., 2004*; *Losonczy and Magee, 2006*; *Major et al., 2008*; *Larkum et al., 2009*; *Lavzin et al., 2012*; *Xu et al., 2012*). Such plateaus require that nearby spines are coactivated, but the spatial requirement is somewhat loose as even single dendritic branches have been proposed to act as computational units (*Branco and Häusser, 2010*; *Losonczy and Magee, 2006*). Nevertheless, multiple so-called hotspots, preferentially responsive to different input values or features, are known to form with close dendritic proximity (*Jia et al., 2010*; *Chen et al., 2011*; *Varga et al., 2011*). Such functional synaptic clusters are present in multiple species, developmental stages, and brain regions (*Kleindienst et al., 2011*; *Takahashi et al., 2012*; *Winnubst et al., 2015*; *Wilson et al., 2016*; *Iacaruso et al., 2017*; *Scholl et al., 2017*; *Niculescu et al., 2018*; *Kerlin et al., 2019*; *Ju et al., 2020*). Hence, multiple features are commonly clustered in a single dendritic branch, indicating that this could be the neural substrate where combinations of simple features into more complex items occur.

Combinations of features in dendritic branches further provide single neurons with the possibility to solve linearly non-separable tasks, such as the nonlinear feature binding problem (NFBP) (*Tran-Van-Minh et al., 2015*; *Gidon et al., 2020*). In its most basic form, the NFBP involves discriminating between two groups of feature combinations. This problem is nonlinear because the neuron must learn to respond only to specific feature combinations, even though all features contribute equally in terms of synaptic input. A commonly used example involves two different shapes combined with two different colors, resulting in four total combinations. Out of these, the neuron should respond only to two specific feature combinations (exemplified in *Figure 1A and B*).

As a task, the NFBP is relevant to brain regions which perform integration of multimodal input signals, or signals representing different features of the same modality (*Roskies, 1999*). It is usually illustrated with examples from the visual system, as in *Figure 1A*; *Roskies, 1999*; *von der Malsburg, 1999*; *Tran-Van-Minh et al., 2015*. A region that integrates multimodal inputs, such as sensory information and motor-related signals, is the input nucleus of the basal ganglia, the striatum (*Reig and Silberberg, 2014*; *Johansson and Silberberg, 2020*), and this system will be used in the present modeling study. Here, however, we will continue to illustrate the NFBP with the more intuitive features borrowed from the visual field, although for the dorsal striatum these features would rather map onto different sensory- and motor-related features. Plateau potentials and some clustering of input have been demonstrated in striatal projection neurons (SPNs) (*Plotkin et al., 2011*; *Oikonomou et al., 2014*; *Du et al., 2017*; *Hwang et al., 2022*; *Day et al., 2024*; *Sanabria et al., 2024*).

In addition to integrating converging input from the cortex and the thalamus, the striatum is densely innervated by midbrain dopaminergic neurons which carry information about rewarding stimuli (*Schultz, 2007*; *Matsuda et al., 2009*; *Surmeier et al., 2010*). As such, the striatum is thought to be an important site of reward learning, associating actions with outcomes based on neuromodulatory cues. In this classical framework, peaks in dopamine signify rewarding outcomes and pauses in dopamine represent the omission of expected rewards (*Schultz et al., 1997*). Dopamine signals further control the synaptic plasticity of corticostriatal synapses on the SPNs (*Figure 1C*). In direct pathway SPNs (dSPNs) expressing the D1 receptor, a dopamine peak together with significant calcium influx through NMDA receptors triggers synaptic strengthening (long-term potentiation [LTP]). Conversely, when little or no dopamine is bound to the D1 receptors, as during a dopamine pause, and there is significant calcium influx through L-type calcium channels, synaptic weakening occurs (long-term depression [LTD], see *Figure 1D*; *Shen et al., 2008*; *Fino et al., 2010*; *Plotkin et al., 2013*).

The ability to undergo LTP or LTD is itself regulated (*Huang et al., 1992*), a concept termed metaplasticity (*Abraham and Bear, 1996*). Metaplasticity refers to changes in synaptic plasticity driven by prior synaptic activity (*Frey et al., 1995*) or by neuromodulators (*Moody et al., 1999*), effectively making plasticity itself adaptable. Metaplasticity can further regulate synaptic physiology to shape future plasticity without directly altering synaptic efficacy, acting as a homeostatic mechanism to keep synapses within an optimal dynamic range (*Abraham, 2008*). Previous theoretical studies have

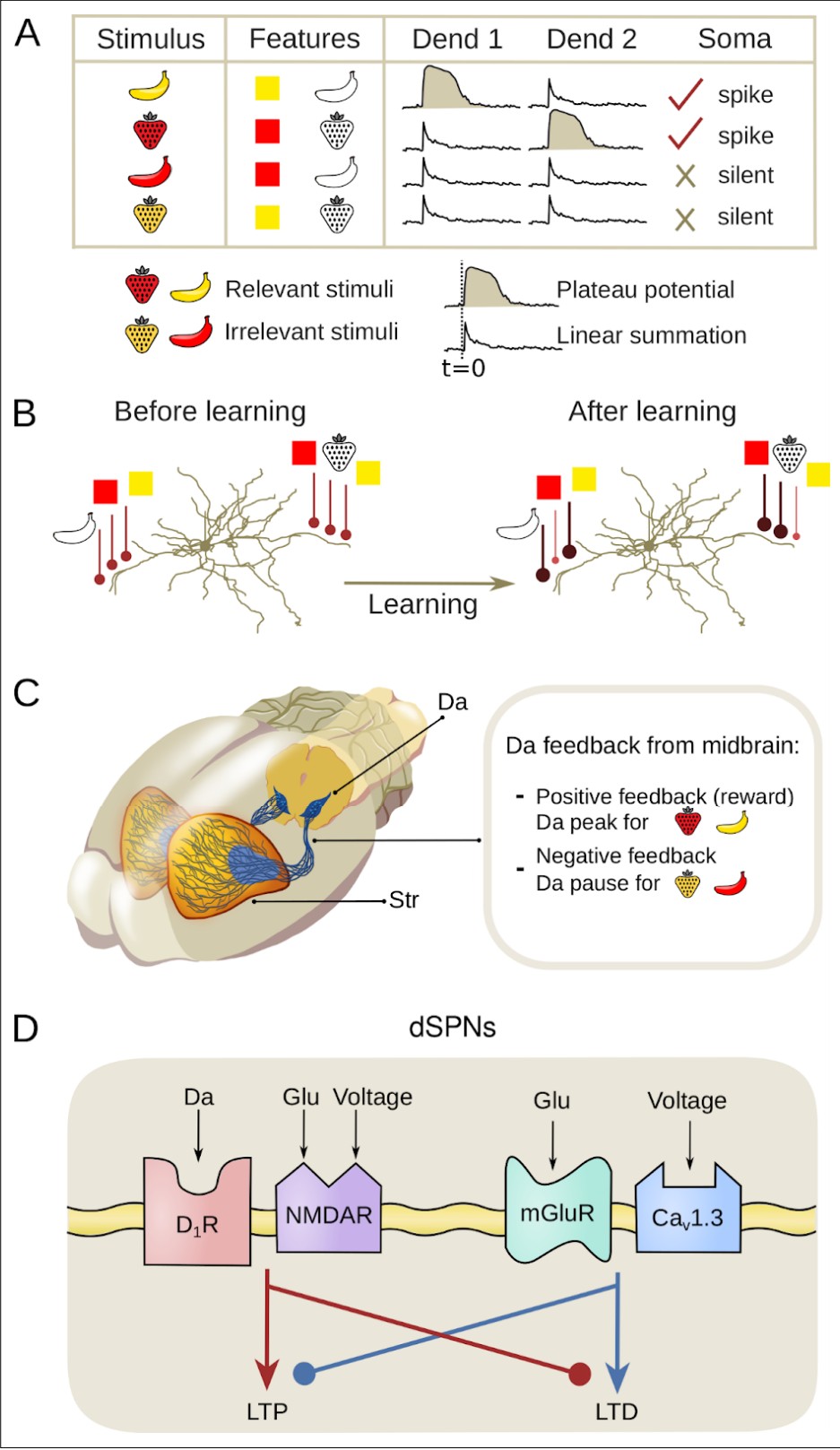

**Figure 1.** Learning mechanisms in direct-pathway striatal projection neurons (dSPNs) for the nonlinear feature binding problem (NFBP). (**A**) Inputs and assumed supralinearity that could solve the NFBP: The NFBP is represented with an example from visual feature binding. In the simplest form of the NFBP, a stimulus has two features, here shape and form, each with two possible values: strawberry and banana, and red and yellow,

*Figure 1 continued on next page*

*Figure 1 continued*

respectively. In the NFBP, the neuron should learn to respond by spiking to two of the feature combinations, representing the relevant stimuli (red strawberry and yellow banana), while remaining silent for the other two feature combinations which represent the irrelevant stimuli (yellow strawberry and red banana). Assuming that each feature is represented with locally clustered synapses, a solution of the NFBP can be achieved when the co-active clusters on a single dendrite, corresponding to a relevant stimulus, evoke a plateau potential, thus superlinearly exciting the soma. Conversely, co-activation of synaptic clusters for the irrelevant combinations should not evoke plateau potentials. (**B**) Synaptic clustering in dendrites: Illustration of how synaptic plasticity in SPNs may contribute to solving the NFBP for a pre-existing arrangement of synaptic clusters on two dendrites. A plasticity rule that strengthens only synaptic clusters representing relevant feature combinations so that they produce robust supralinear responses, while weakening synapses activated by irrelevant feature combinations, could solve the NFBP. (**C**) Dopamine (Da) feedback: Dopaminergic inputs from the midbrain to the striatum (Str) guide the learning process, differentiating between positive feedback for relevant stimuli and negative feedback for irrelevant stimuli. Positive feedback, represented by a dopamine peak, is necessary for long-term potentiation (LTP), and negative feedback, represented by a dopamine pause, is necessary for long-term depression (LTD). (**D**) Signaling pathways underlying synaptic plasticity in dSPNs: illustrations of signaling components at the corticostriatal synapse that modify synaptic strength (redrawn from *Shen et al., 2008*). NMDA calcium influx, followed by stimulation of D1 dopamine receptors (D1Rs), triggers LTP (while inhibiting the LTD cascade). L-type calcium influx and activation of metabotropic glutamate receptors (mGluRs) when D1Rs are free of dopamine triggers LTD (while counteracting the LTP cascade).

demonstrated the essential role of metaplasticity in maintaining stability in synaptic weight distributions (*Bienenstock et al., 1982*; *Fusi et al., 2005*; *Clopath et al., 2010*; *Benna and Fusi, 2016*; *Zenke and Gerstner, 2017*).

If dopamine peaks are associated with the relevant feature combinations in the NFBP and dopamine pauses with the irrelevant ones, they trigger LTP in synapses representing the relevant feature combinations and LTD in those representing irrelevant combinations. If, after learning, the relevant feature combinations have strong enough synapses so they can evoke plateau potentials, while the irrelevant feature combinations have weak enough synapses so they don't evoke plateaus, the outcome of this learning process should be a synaptic arrangement that could solve the NFBP (*Figure 1B*; *Tran-Van-Minh et al., 2015*). In line with this, it has been demonstrated that the NFBP can be solved in abstract neuron models where the soma and dendrites are represented by single electrical compartments and where neuronal firing and plateau potentials are phenomenologically represented by instantaneous firing rate functions (*Legenstein and Maass, 2011*; *Schiess et al., 2016*). Good performance on the NFBP has also been demonstrated with biologically detailed models (*Bicknell and Häusser, 2021*). This solution used a multicompartmental model of a single pyramidal neuron, including both excitatory and inhibitory synapses and supralinear NMDA depolarizations. Synapses representing different features were randomly dispersed throughout the dendrites, and a phenomenological precalculated learning rule – dependent on somatic spike timing and high local dendritic voltage – was used to optimize the strength of the synapses. The solution did, however, depend on a form of supervised learning as somatic current injections were used to raise the spiking probability of the relevant feature combinations.

In this study, we ask whether – and under what conditions – the theoretical solution to the NFBP can be achieved in a biophysically detailed model of an SPN using only local, biologically-grounded learning rules. We frame this paper around two questions: First, can a single dSPN equipped with only calcium- and dopamine-dependent excitatory plasticity solve the NFBP when the relevant features are pre-clustered on one dendritic branch? Second, if that mechanism is insufficient – as with randomly distributed or very distal inputs – does adding inhibitory plasticity restore plateau-based nonlinear computation and spiking?

To answer these questions, we adopt an approach that relies on the following key mechanisms:

**A local learning rule**: We develop a learning rule driven by local calcium dynamics in the synapse and by reward signals from the neuromodulator dopamine. This plasticity rule is based on the known synaptic machinery for triggering LTP or LTD at the corticostriatal synapse onto dSPNs (*Shen et al., 2008*). Importantly, the rule does not rely on supervised learning paradigms, and no separate training and testing phase is required.

**Robust dendritic nonlinearities**: According to *Tran-Van-Minh et al., 2015*, sufficient supralinear integration is needed to ensure that, e.g., two inputs (one feature combination in the NFBP, *Figure 1A*) on the same dendrite generate greater somatic depolarization than if those inputs were distributed across different dendrites. To accomplish this, we generate dendritic plateau potentials using the approach of *Trpevski et al., 2023*.

**Metaplasticity**: Our simulations demonstrate that metaplasticity is necessary for synaptic weights to remain stable and within physiologically realistic ranges, regardless of their initial values.

We first demonstrate the effectiveness of the proposed learning rule under the assumption of pre-existing clustered synapses for each individual feature, as suggested by *Tran-Van-Minh et al., 2015*. These clustered synapses are trained to a degree where they can reliably evoke robust plateau potentials for the relevant feature combinations required to solve the NFBP, while synapses representing irrelevant features are weakened (illustrated in *Figure 1B*).

We then extend the analysis by applying the learning rule to more randomly distributed synapses, which initially exhibit minimal local supralinear integration. However, when incorporating the assumption that branch-specific plasticity mechanisms are at play, supralinear integration emerges within distinct dendritic branches. This suggests that branch-specific plasticity could play a critical role in enabling single neurons to solve nonlinear problems. Furthermore, we explore an activity-dependent rule for GABAergic plasticity and demonstrate its potential importance in shaping dendritic nonlinearities. This mechanism may thus further enhance computational capabilities by refining and stabilizing the integration of inputs across dendritic branches.

Although brain systems like the striatum, which integrate multimodal inputs, somehow solve nonlinear problems at the network or systems level, it remains unclear whether individual neurons in the brain regularly solve the NFBP. Our investigation suggests, however, that single SPNs possess the computational capacity to address linearly non-separable tasks. This is achieved by leveraging the organism's performance feedback, represented by dopamine peaks (success) and dopamine pauses (failure), in combination with their ability to generate dendritic plateau potentials. Since the mechanisms used in the rule are general to the brain, this capability may also extend to other projection neurons capable of producing dendritic plateaus, such as pyramidal neurons. However, the specific feedback mechanisms, represented here by dopamine, would need to be associated with alternative neuromodulatory signals depending on the type of neuron and synapse involved.

## Results

### Characterization of the dendritic nonlinearities in the model

The nonlinear sigmoidal voltage sensitivity of NMDA receptors is a crucial element for forming dendritic plateau potentials. We used a model to generate plateau potentials first presented in *Gao et al., 2021*, and adjusted to SPNs (*Trpevski et al., 2023*). To produce robust all-or-none plateau potentials, the model includes glutamate spillover from the synaptic cleft that activates extrasynaptic NMDA receptors. Glutamate spillover occurs when the total synaptic weight of the nearby activated synapses reaches a threshold value (see Materials and methods). The threshold value here is set to be equivalent to the total weight of eight clustered synapses with weights of 0.25 each (weights of 0.25 correspond to a maximal conductance of 0.625 nS). This produces dendritic plateaus which, compared to *Trpevski et al., 2023*, are not as large and all-or-none, thus situating the dendrites in the softer 'boosting' regime, which is likely common under physiological conditions.

*Figure 2A* shows the somatic membrane potential following synaptic activation of a cluster of synapses of increasing size and the corresponding local spine membrane potential, as well as the NMDA and L-type calcium accumulated in a single spine (averaged over all spines in the cluster). A plateau potential is generated when a critical level of total NMDA conductance in a dendritic segment is reached (accomplished here by the addition of more synapses in a cluster, eventually leading to glutamate spillover). Reaching the spillover threshold produces a sudden and prolonged increase in NMDA conductance, caused by the activation of extrasynaptic NMDA receptors (where the clearance of glutamate is assumed to be slower).

The synaptic input to the neuron is provided through the activation of a cluster of synapses at an example location in *Figure 2B* and gives the voltage and calcium responses as exemplified in *Figure 2A*. *Figure 2C* shows the maximal amplitudes of the somatic and spine voltage together with

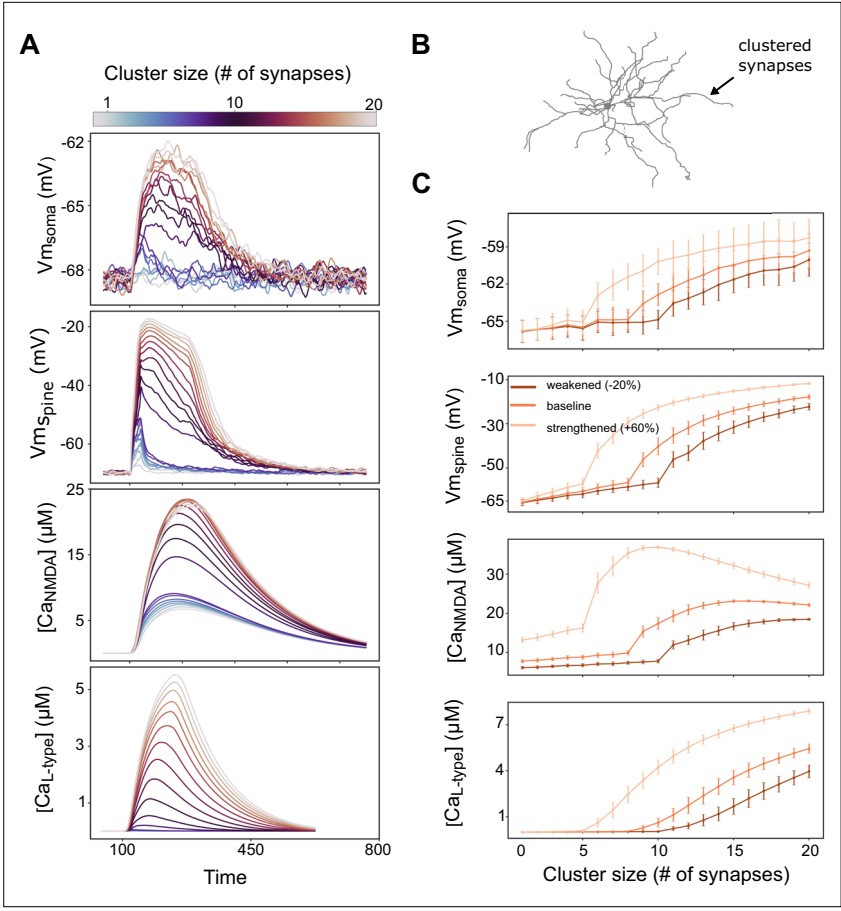

**Figure 2.** Characterization of dendritic plateau potentials in the model. (**A**) Somatic voltage, spine voltage, NMDA calcium, and L-type calcium evoked by a cluster varying in size from 1 to 20 synapses. A plateau potential is evoked when glutamate spillover is activated, here triggered when 8 synapses with a weight of 0.25 each are coactivated (corresponding to the 'baseline' weights in C). (**B**) Schematic of the neuron morphology with an arrow indicating an example location of a stimulated dendritic branch. (**C**) The mean maximal amplitude, with standard deviation shown in bars, of the measures in A averaged over 10 different dendrites and 10 trials per dendrite (for a total of n = 100 trials). The curves represent clusters with different synaptic weights (conductances): baseline – 0.25 (0.625 nS), strengthened – 0.4 (1 nS), and weakened – 0.2 (0.5 nS). Synaptic background noise is used in all simulations to elevate the membrane potential to ranges seen in vivo (***Reig and Silberberg, 2014***).

the amplitudes of NMDA and L-type calcium signals in the dendritic spines, averaged over 10 trials and over 10 dendrites. The 'baseline' results in *Figure 2C* are within the range of the initial synaptic weights of excitatory synapses in all remaining figures in the article, and thus illustrate a possible initial situation before learning. Stronger and weaker synapses require a smaller and a larger cluster to trigger a plateau potential, respectively (*Figure 2C*, spine voltage panel). In the simulation using 'strengthened' synapses, the synaptic weights in the cluster are 60% greater than in the 'baseline' case and therefore need fewer synapses to trigger plateau potentials. In contrast, with weaker synapses, where the weights are 20% smaller than the 'baseline' case, more synapses are needed to evoke a plateau.

To summarize, the dSPN model exhibits the dendritic nonlinearities required for solving the NFBP. Further, clusters of strengthened synapses can reliably generate robust plateau potentials, which produce a long-lasting somatic depolarization and thereby increase the likelihood for somatic spiking (in accordance with *Du et al., 2017*, and also illustrated in *Trpevski et al., 2023*). Conversely, clusters of weakened synapses will most likely not generate plateau potentials, and thus the neuron will spike with much lower probability following activation of such a cluster.

## Characterization of the synaptic plasticity rule

To characterize the learning rule, we started with a simple setup where three features, each illustrating either a color or a shape, were distributed onto two dendritic branches. The setup was such that one relevant and one irrelevant feature combination were represented in each dendrite and the relevant–irrelevant feature combination was unique to each dendrite (see illustration in *Figure 3A*). Each feature was represented with 5 synapses, and we start with the assumption that those synapses are already organized in pre-existing clusters. In addition to background synaptic noise inputs, 108 randomly distributed glutamatergic synapses were also added. These synapses were activated together with all four stimuli, i.e., they were feature-unspecific. SPNs have a very hyperpolarized resting potential, and the additional feature-unspecific synapses allowed the neurons to spike often enough in the beginning of the learning process so that a dopamine feedback signal would be elicited and trigger learning in the activated synapses.

The learning rule modifies synaptic weights based on local calcium levels at each synapse, employing distinct plasticity kernels for LTP and LTD, each associated with different calcium sources – NMDA and L-type channels, respectively (*Figure 1*; schematic in *Figure 3B*; see also Materials and methods). The LTD kernel was governed by a calcium threshold, above which the change of synaptic weight was proportional to the amplitude of L-type calcium. That is, higher calcium levels trigger a larger reduction of weights than low concentrations. Conversely, the LTP kernel was represented by a bell-shaped function over a range of NMDA calcium concentrations, allowing LTP in the presence of dopamine when calcium levels fell within this range, while calcium outside this range did not elicit LTP (see Materials and methods).

An additional feature of the learning rule was metaplasticity (*Abraham and Bear, 1996*), which dynamically adjusted the calcium dependence of the LTP kernel. That is, the LTP kernel's calcium dependence was adjusted over time (see *Figure 3B–D*). Metaplasticity enabled previously weak synapses, which were repeatedly co-activated with rewards, to eventually participate in LTP, as their calcium threshold was gradually lowered when the LTP kernel was shifted. The same mechanism also prevented excessive strengthening of already potent synapses, as shifting the LTP kernel downward also reduced the potentiation of synapses with large calcium transients. This behavior of the LTP kernel ensured stability in synaptic weights and allowed for continuous refinement of synaptic efficacy based on activation pattern and presence/absence of dopamine.

Initially, the synaptic weight of each synapse was set to a value chosen uniformly at random from a range of 0.25 ± 0.05 (corresponding to conductances of 0.625 ± 0.125 nS). Initially, the synapses therefore experienced different NMDA and L-type calcium levels following activation, and hence their weights were also updated differently based on where the calcium level fell within their individual plasticity kernels.

During training, the neuron model was activated with a sequence of 1400 feature combinations, including equal amounts of relevant (i.e. 'red strawberry' and 'yellow banana') and irrelevant feature combinations (i.e. 'yellow strawberry' and 'red banana'). Thus, dopamine peaks and dopamine pauses occurred equally often at the beginning of the learning phase. When the neuron spiked for the relevant feature combinations, dopamine rewards were delivered, triggering LTP in the active synapses with NMDA calcium levels within the LTP kernel. Conversely, spiking for the irrelevant feature combinations elicited a dopamine pause as feedback, triggering LTD as a function of L-type calcium. Initially, all four stimuli – 'yellow banana' and 'red banana' in dendrite 1, and 'red strawberry' and 'yellow strawberry' in dendrite 2 – elicited robust supralinear responses, as they together reached the threshold for glutamate spillover in the model (*Figure 3E*, gray lines). After learning, however, the neuron could differentiate between the two sets of stimuli – the relevant feature combinations associated with a reward continued to evoke a plateau potential and elicit somatic spiking, while in contrast, the neuron's response to irrelevant feature combinations was notably decreased (*Figure 3E*, black lines).

*Figure 3C* shows the evolution of synaptic conductances during the learning process for dendrite 1 (dendrite 2 is not shown, but has a similar behavior for relevant and irrelevant stimuli). The synapses representing the relevant feature combination in this dendrite ('yellow' and 'banana') are typically strengthened, eventually encoding this stimulus robustly. Conversely, the synapses for the feature 'red', activated during the irrelevant feature combination ('red banana'), are all weakened, making the dendrite only weakly responsive to this stimulus following learning. Note that during the learning process, LTD could also occur in some synapses representing 'yellow' and 'banana' because these

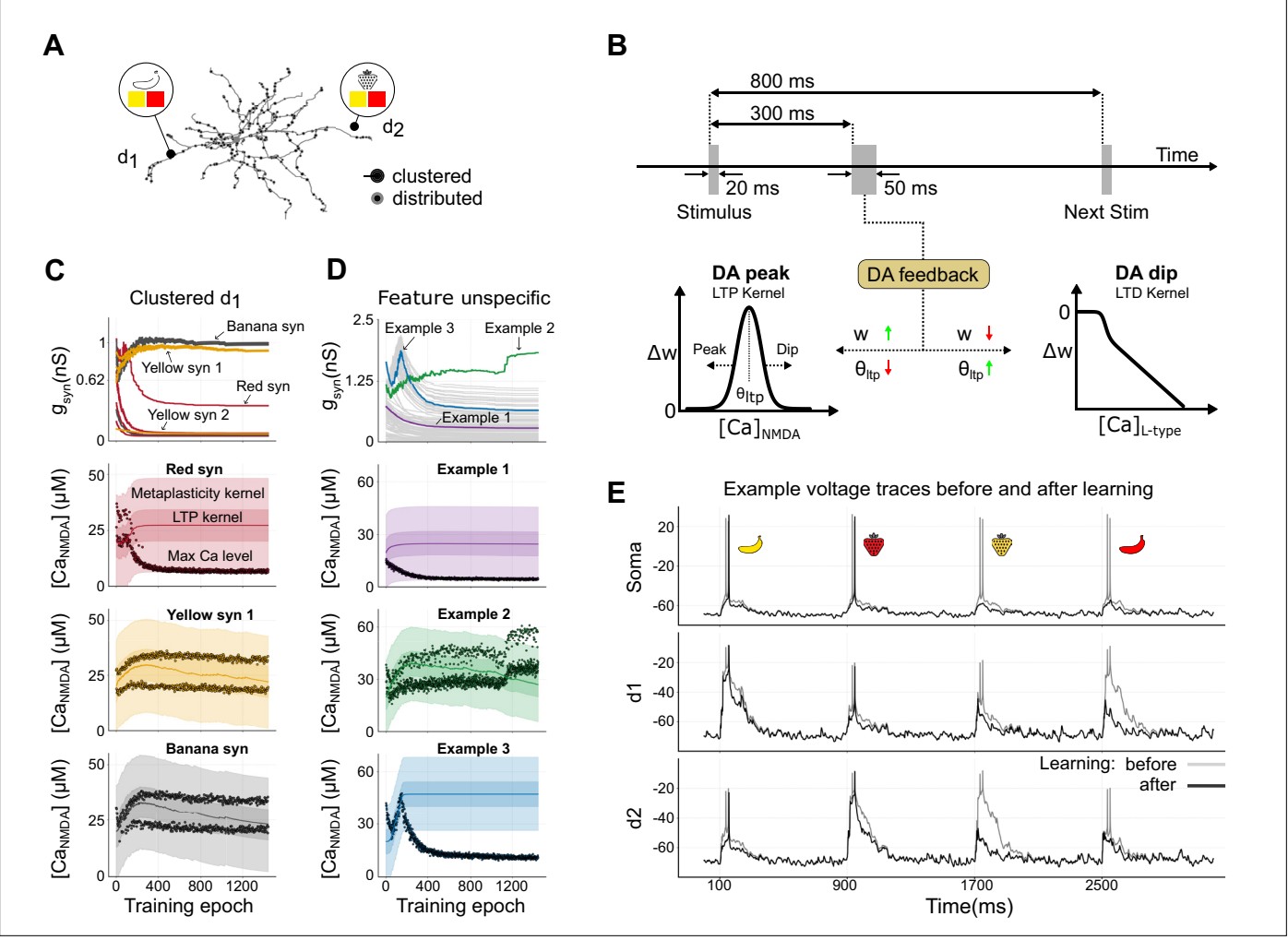

**Figure 3.** Example of setup and learning-induced synaptic plasticity. (**A**) Input configuration. The panel shows how the four stimulus features are distributed across two dendrites (d1, d2). Each dendrite contains pre-existing synaptic clusters for three features (black circles) and distributed, feature-unspecific synapses (shown in gray). (**B**) Stimulation timeline and plasticity schematic. The top diagram depicts a 20 ms stimulus followed by a 50 ms dopaminergic feedback pulse delivered with a 300 ms delay, only if the neuron spikes. Successive stimuli are separated by 800 ms to let calcium return to baseline. Lower panels: a dopamine peak (left) gates long-term potentiation (LTP) – synapses whose $[Ca]_{NMDA}$ falls within the bell-shaped window are potentiated ($w \uparrow$, green) while the kernel midpoint $\theta_{LTP}$ shifts downward ($\downarrow$, red); a dopamine pause (right) gates long-term depression (LTD) – the degree of depression scales with $[Ca]_{L\text{-type}}$ ($w \downarrow$, red) and $\theta_{LTP}$ shifts upward ($\uparrow$, green). (**C, D**) Evolution of synaptic conductances during learning (top row) and examples of peak $Ca^{2+}$ levels (black dots) alongside kernel dynamics in single synapses (three lower rows). Panels in (**C**) depict clustered synapses in dendrite 1 (d1), where 'yellow' and 'banana' generally undergo LTP, while 'red' undergoes LTD. See *Figure 3—video 1* for an animation of panel C showing the first 400 training stimuli of the learning sequence. Panels in (**D**) show distributed, feature-unspecific synapses. Among these, Example 1 and Example 3 traces represent synapses that are weakened, while Example 2 trace exemplifies a synapse near one of the clusters that, by chance, achieves a sufficiently high local NMDA calcium level for LTP to dominate. The initial synaptic conductances are drawn from a normal distribution with a mean of 0.625 nS and a standard deviation of 0.125 nS. The solid line represents the midpoint of the kernels, where LTP is strongest. 'Max' indicates the peak NMDA calcium during a single stimulus. The darker shaded regions represent the LTP kernel, and the lighter shaded ones show the wider metaplasticity kernel. (**E**) Example voltage in the soma and the middle of dendrite 1 (d1) and dendrite 2 (d2) before and after learning. Each dendrite here stops responding to the respective irrelevant stimuli during learning.

The online version of this article includes the following video and figure supplement(s) for figure 3:

**Figure supplement 1.** Learning-induced synaptic plasticity with metaplasticity turned off.

**Figure 3—video 1.** Animated rendering of the first 400 training stimuli (~400 epochs) showing frame-by-frame changes in conductance and calcium for the clustered synapses shown in panel C.

https://elifesciences.org/articles/97274/figures#fig3video1

features were also part of the irrelevant stimuli ('yellow strawberry' and 'red banana', respectively). As a result, a small number of these synapses, in particular those whose initial weights are low, have been weakened and are not recruited in the clusters to encode the 'yellow banana' stimulus. This means that our learning rule tends to stabilize the number of synapses that are needed to perform the task, but not necessarily all the synapses carrying the relevant features (as some by chance may be outside of the plastic region of the LTP kernel). Depending on the initial local calcium response in the synapse to relevant and irrelevant stimuli, individual synapses might either be preferentially recruited into the LTP or LTD process. An animated rendering of the first 400 training epochs for this panel, illustrating the frame-by-frame evolution of conductance and calcium in the example with clustered synapses, is provided in *Figure 3—video 1*.

The evolution of synaptic weights for clustered and feature-unspecific synapses is illustrated in the top panels of *Figure 3C and D*. Correspondingly, the calcium dynamics for three color-coded synapses in each case (indicated with arrows in the top panels) are shown in the bottom panels, demonstrating how the dynamics evolve under the influence of LTP, LTD, and metaplasticity. This highlights the process by which synapses stabilize at specific conductances/weights during learning.

The red synapses (exemplified in *Figure 3C*, second row) represent an irrelevant feature combination ('red banana'), and initially LTD is frequent due to repeated dopamine pauses. These pauses shift the LTP and metaplasticity kernels toward higher calcium levels, moving the red synapses outside the range for LTP. This shift ensures that the red synapses weaken over time, preventing them from encoding irrelevant stimuli and reducing the dendrite's response to these inputs. In setups lacking metaplasticity, this does not happen, leading to an inability of the neuron to separate the response to relevant and irrelevant feature combinations (*Figure 3—figure supplement 1*, red synapse).

Most synapses representing the relevant feature combination ('yellow banana', exemplified in *Figure 3C*, bottom rows) initially undergo LTP as repeated dopamine rewards strengthen them. However, synapses with weak initial weights may fail to potentiate further due to the slow adaptation of the LTP kernel and their inability to reach the required calcium levels. Specifically, for the synapses representing yellow, 'Yellow syn 1' exhibits an increase in synaptic conductance, whereas 'Yellow syn 2' does not undergo potentiation (see arrows in *Figure 3C*, top).

Feature-unspecific synapses, like Example 1 and 3 in *Figure 3D*, typically weaken over time. These synapses are co-activated with irrelevant stimuli and experience frequent dopamine pauses, causing the metaplasticity kernel to shift upward and away from the calcium levels required for LTP. As a result, LTD dominates in these synapses, leading to their gradual weakening. Initially, LTP and LTD typically occur equally often, but as learning progresses, both the frequency of dopamine pauses and thus LTD decrease, allowing the weights to stabilize over time.

In contrast, Example 2, though feature-unspecific, behaves differently due to its proximity to a cluster. It benefits from high local voltage and calcium during relevant stimuli (e.g. 'yellow banana'), keeping calcium levels within the high end of the LTP kernel. As LTP dominates in this synapse, the LTP kernel gradually shifts down toward lower calcium concentrations. Eventually, this causes the calcium level associated with the other relevant stimulus ('red strawberry') to also fall within the plastic region of the LTP kernel, and thereby further strengthening the synapse.

In summary, synaptic weights are dynamically regulated by the interplay of LTD and LTP, and the latter is modulated by the metaplasticity kernel. This process ensures that synapses adjust appropriately based on their relevance to feature combinations, their functional roles and initial conditions.

## Clustering enables excitatory-only learning, but overlapping features make performance less reliable

After demonstrating that the SPN can differentiate between relevant and irrelevant stimuli in the simplified example in *Figure 3*, we generalized this setup by varying the configuration of the four features across the two dendrites and recorded the SPN's performance on the NFBP as learning progressed. We only used feature configurations where both relevant feature combinations were present, with at least one relevant feature on each dendrite. This ensures sufficient innervation to potentially solve the NFBP (for an illustration, see *Figure 4A*). The SPN's performance was assessed during training to determine whether it spiked for relevant feature combinations and remained silent for irrelevant ones. Performance of 100% indicates the SPN spikes exclusively for relevant stimuli, while 50% typically indicates one of two scenarios: (i) the SPN spikes for all four stimuli, or (ii) it remains silent for all four.

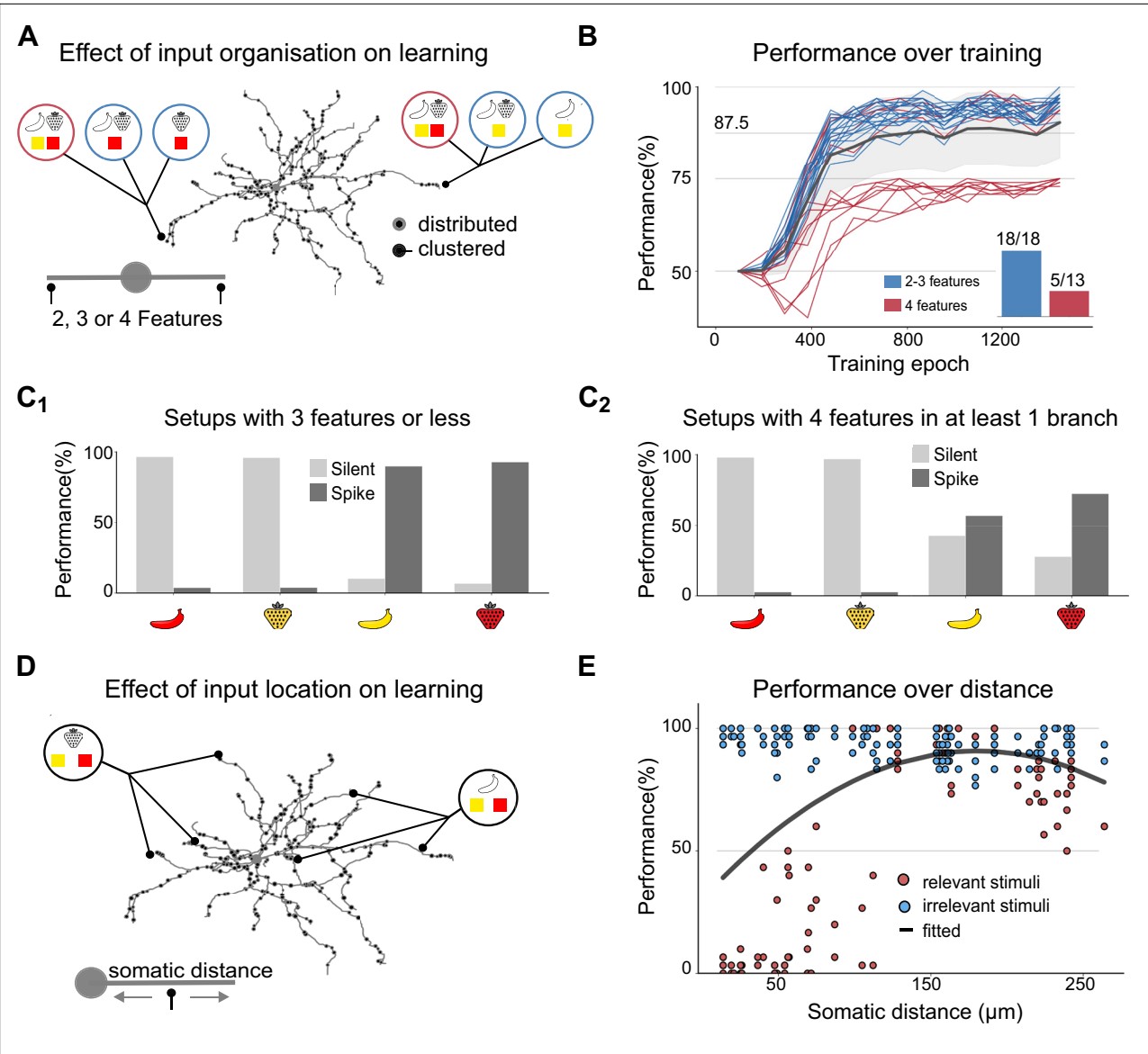

**Figure 4.** Impact of input configurations and synaptic cluster locations on nonlinear feature binding problem (NFBP) learning performance. (**A**) Illustration of creating different input configurations where two, three (light blue circles), or four features (deep red circles) are placed in two dendritic locations. (**B**) Performance trajectories for all 31 unique input configurations over training epochs, shown for one particular pair of dendrites. Each colored line represents the learning performance of a single configuration, with blue traces indicating setups where no dendrite receives more than three features and red traces representing setups where at least one dendrite contains all four features. The black line and gray shaded area represent the mean and standard deviation over all trials (n = 31). The inset shows the number of configurations in each group that reached the NFBP learning criterion: 18/18 for the blue group and 5/13 for the deep red group. (**C**) Performance on the last 160 stimuli for the two groups of configurations: (**C1**) setups where each dendrite has at most three features and (**C2**) setups where at least one dendrite contains all four features. (**D, E**) Performance in a three-feature configuration as a function of cluster location. (**D**) illustrates the distribution of synaptic clusters across dendritic locations, and (**E**) shows performance over the last 160 stimulus presentations as a function of somatic distance for these configurations. A total of 60 unique dendritic arrangements were tested, with synapse clusters randomly assigned to different dendritic locations. The solid line represents a quadratic fit to the data.

A performance of 75% reflects intermediate spiking behavior, such as spiking for one relevant combination while remaining silent for the other three, or spiking for three combinations and being silent for one irrelevant combination. The NFBP is considered solved when performance exceeds 87.5%, which, for example, occurs when the SPN consistently spikes for one relevant combination and spikes at least half the time for the other relevant combination while ignoring irrelevant stimuli.

In cases where both dendrites receive two or three features, ensuring that each dendrite has at least one relevant combination, the mean performance exceeds 90%, indicating that both relevant

combinations are learned (*Figure 4B*, blue traces). However, when at least one dendrite is innervated by all four features, learning becomes significantly more challenging. The dendrite with four features struggles to resolve the competition between feature combinations, making it difficult to encode a single relevant combination (*Figure 4B*, deep red traces). However, in some cases, depending on the order of stimuli during training, both relevant combinations can still be learned, solving the NFBP. Here, 5 out of 13 setups in which one dendrite received all four features successfully solved the task (*Figure 4B*, inset).

*Figure 4C* provides a detailed view of performance during the last 160 training examples for each of the four stimuli. *Figure 4C1* corresponds to the blue traces in *Figure 4B*, where the NFBP is considered solved. *Figure 4C2* corresponds to the deep red traces in *Figure 4B*, where the NFBP is generally not solved, as relevant stimuli elicit spikes only about 75% of the time. These results suggest that when one dendrite is innervated by all four features, the competition between feature combinations creates a bottleneck for learning. In cases where learning fails, the dendrite with four features often either fails to encode any combination due to excessive LTD or instead encodes the same combination already represented in the other dendrite. Addressing this challenge may require mechanisms to 'break symmetry' during learning, as noted in previous studies using abstract neuron models. Symmetry-breaking strategies, such as amplifying or attenuating dendritic nonlinearities as they propagate to the soma can enable successful differentiation of feature combinations (e.g. branch plasticity; *Legenstein and Maass, 2011*).

## Optimal learning is achieved at intermediate distances from soma through excitatory plasticity

We also investigated the impact of synaptic positioning on learning (*Figure 4D*) when using the same input setup as in *Figure 3*, but varying dendritic locations. Our results predict that the best performance on the NFBP is obtained with synaptic clusters positioned at intermediate somatic distances from the soma (*Figure 4E*). From *Figure 4E*, one can infer that after learning, the proximal synapses have all been decreased to such an extent that the neuron has stopped spiking for all four combinations (as the total performance is around 50% and relevant feature combinations are at 0%). For successively more distal synapses, the performance increases and then slightly decreases for the most distal clusters that sometimes fail to evoke somatic spiking for the correct feature combinations (red dots). This result can be conceptually explained in the following way. The electrotonic properties of dendrites dictate that synapses near the soma, in the most proximal regions, are less capable of supralinear input integration underlying plateau potentials (*Du et al., 2017*). This is due to the soma acting as a current sink, resulting in smaller localized voltage changes (and hence a lower input resistance in accordance with Ohm's law). Consequently, these synapses cannot easily evoke dendritic nonlinearities necessary for solving the NFBP, and hence the performance with proximal clusters is low. Note that in our simulations, we allow glutamatergic synapses on spines quite close to the soma, although SPN dendritic spines are relatively rare at more proximal distances than 40–50 μm from the soma (*Wilson et al., 1983*).

In contrast, the most distal dendritic regions are electrically more isolated and have a higher local input resistance, enabling larger voltage changes locally and thus also higher local calcium concentrations when synapses are activated in our model. This allows even a small number of active synapses to generate local supralinear NMDA-dependent responses. Such ease of elevating the local calcium seems advantageous; however, this ultimately results in decreased performance on the NFBP for the following reasons. In distal synaptic clusters, excessive spiking for irrelevant stimuli occurs during the early and middle stages of learning. This leads to frequent dopamine pauses (negative feedback), which, in turn, reduce the neuron's ability to spike for relevant combinations later during training. Additionally, while distal synapses can generate plateau potentials and NMDA-dependent responses, these signals naturally attenuate as they propagate toward the soma, sometimes failing to elicit somatic spikes for the relevant stimuli. This combination of excessive negative feedback early on and attenuated distal signals reduces the effectiveness of the feedback loop, making it difficult for the neuron to selectively strengthen synapses associated with relevant stimuli and weaken those tied to irrelevant ones.

In our simulations, the ideal learning zone for NFBP thus lies at an intermediate somatic distance, around 150 μm from the soma, where synapses can effectively contribute to learning the NFBP

(*Figure 4E*). In this zone, synaptic changes are more likely to impact the neuron's firing probability as the dendritic plateau potential at this location causes a larger elevation of the somatic potential, and thus synapses at this distance benefit more from the dopamine feedback loop. Performance in this zone shows variability across dendritic pairs, as also illustrated in *Figure 4E* with red and blue markers, reflecting differences in the strength of local nonlinear responses. Note that the prediction that proximal and very distal synapses are less likely to contribute to the solving of the NFBP doesn't imply they are not important for more 'linear' learning contexts. For instance, if we had trained the neuron to only respond to one single stimulus, such as 'red strawberry', both proximal and very distal synapses representing that stimulus would of course be able to both strengthen or weaken and contribute to spiking of the neuron following learning.

## Inhibitory plasticity can enhance contrast and robustness of dendritic computations

In our initial simulations, we assumed that only the excitatory synapses could undergo plasticity during learning, and we identified two critical observations that highlight possible areas for improvement.

The first is a vulnerability to noise, which resulted in a performance around 90%, as illustrated in *Figure 4B* on one particular pair of dendrites, and in the optimal zone shown in *Figure 4E*. The second is a decrease in performance observed across very distal synapses, as detailed in *Figure 4E*. These findings prompted us to explore the potential role of inhibitory synapses.

We developed a phenomenological inhibitory plasticity rule to enhance the contrast of dendritic nonlinearities (see *Figure 5A*, detailed in the Materials and methods section). This rule is designed to compartmentalize dendrites, ensuring they predominantly respond to excitatory inputs that cause the strongest activation. Such compartmentalization has been experimentally observed in radial oblique and basal dendrites of CA1 pyramidal neurons, where NMDA spikes and plateau potentials are the main forms of dendritic nonlinearity. Co-located inhibitory synapses in these dendrites regulate whether these nonlinear responses are elicited (*Lovett-Barron et al., 2012*; *Milstein et al., 2015*; *Grienberger et al., 2017*).

Unlike excitatory plasticity, which relies on dopaminergic feedback signals, our inhibitory plasticity model follows a rule that passively follows the local calcium level from voltage-gated channels. It reinforces the voltage elevation of the most active excitatory synapses within a dendritic branch by decreasing the inhibition corresponding to the same features there (*Chapman et al., 2022*), and conversely, it strengthens inhibitory synapses for features which generate less excitatory activity (see *Figure 5A*, right panel, illustrating how the rule updates active and inactive inhibitory synapses).

To demonstrate how the inhibitory plasticity rule works, we use the same excitatory synapse setup as in *Figure 4* to which we add four inhibitory synapses near each cluster in the middle of the dendritic branch, representing each of the four features (*Figure 5A*). Thus, a single feature activates both the excitatory and inhibitory synapses. To achieve a level of depolarization and spike probability comparable to that in our excitatory-only setup, we increased the number of feature-unspecific inputs from 108 to 144. Alongside this, we began with low inhibitory synaptic weights. These two changes were needed in order to maintain higher baseline activity in the model, as starting with strong inhibitory weights could excessively suppress excitatory activity, since inhibitory inputs close to clustered synapses can effectively counteract the NMDA-dependent nonlinearities (*Doron et al., 2017*; *Du et al., 2017*; *Dorman et al., 2018*). Here, as in the example in *Figure 3*, the conductances of the excitatory inputs representing 'yellow' and 'banana' increase in dendrite 1, strengthening the 'yellow banana' pairing, while the weights of synapses representing the feature 'red' decrease (*Figure 5E*). Conversely, the inhibitory synapses associated with the 'yellow' and 'banana' features in dendrite 1 are weakened, while those linked to the 'red' and 'strawberry' features in the same dendrite are strengthened (*Figure 5D*, right panel). This behavior of the inhibitory plasticity rule effectively prevents the cell from spiking following activation of the irrelevant stimuli. In this particular example, the 'red banana' combination is strongly inhibited, thereby effectively compartmentalizing dendrite 1 to be responsive to 'yellow banana' (since in this case no excitatory 'strawberry' input is present).

We also show the dynamics of peak calcium levels associated with both excitatory and inhibitory synapses for each task (*Figure 5D*, left panel, and *Figure 5E*). For excitatory synapses, the patterns in peak NMDA calcium levels behave as in the example in *Figure 3C* (compare to *Figure 5E*). For inhibitory synapses, the peak calcium levels arising from voltage-gated calcium channels further exemplify

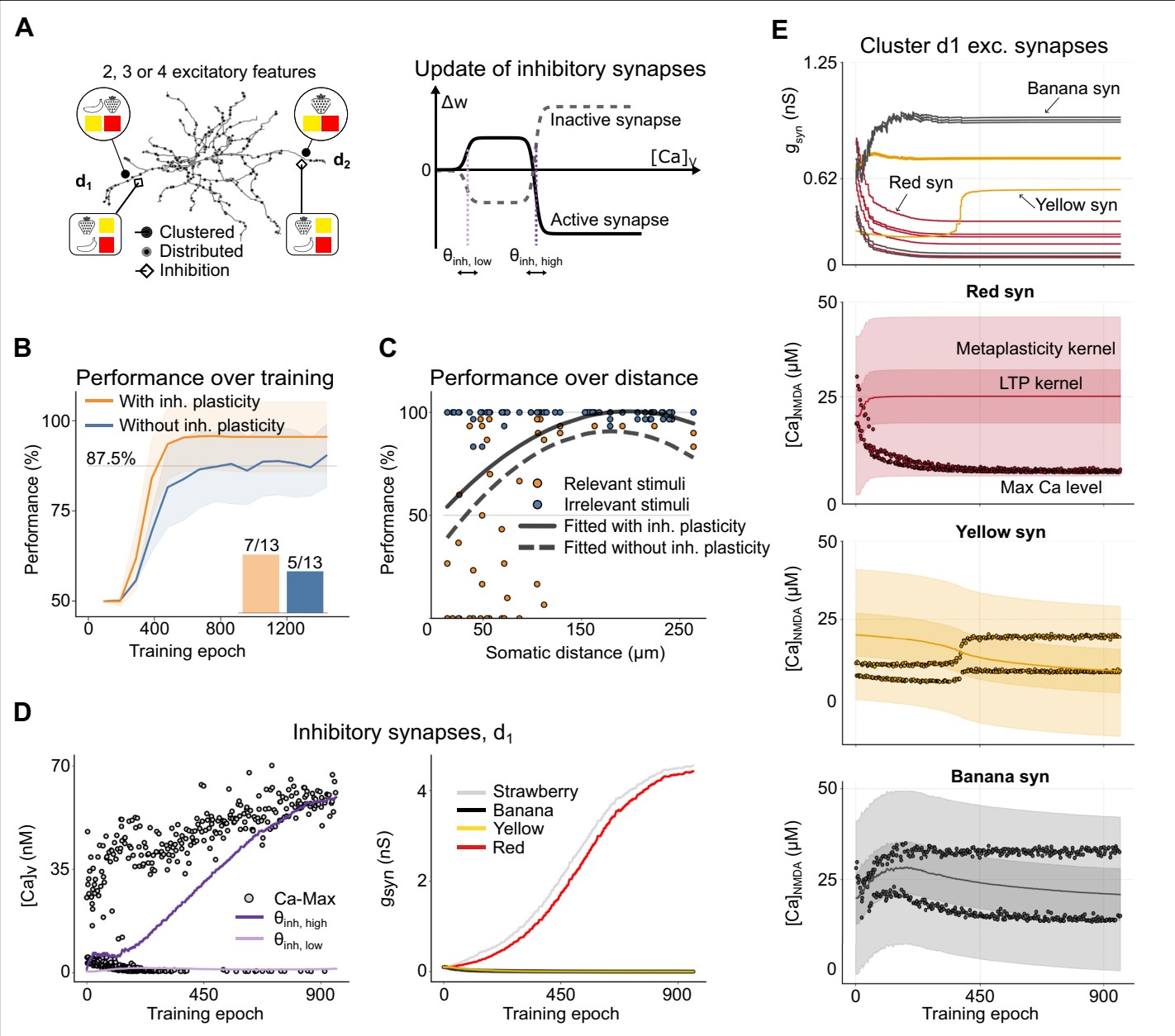

**Figure 5.** Effects of inhibitory plasticity on performance. (**A**) Dendritic input configuration with inhibitory synapses added. The setup of the excitatory and feature-unspecific synapses is the same as in *Figure 4A*. Plastic inhibitory synaptic connections for each of the four features are added in both dendrites, with one synapse per feature. To the right is a schematic of the inhibitory plasticity rule. Active inhibitory synapses strengthen at lower calcium levels and weaken when calcium is high, whereas inactive inhibitory synapses follow the opposite pattern. (**B**) displays average performance for 31 unique input configurations of 2, 3, or 4 features on two dendrites as a comparison between the setup with (orange) and without inhibitory plasticity (blue). Shaded areas show standard deviation (n = 31). The bar plot inset shows the number of configurations, with 4 features in at least one dendrite, that successfully solved the nonlinear feature binding problem (NFBP) with and without inhibitory plasticity. (**C**) Performance over the last 160 stimulus presentations as a function of somatic distance of the synaptic clusters for the input configuration in *Figure 4D*, with added inhibition. A total of 60 unique dendritic arrangements were tested, with synapse clusters randomly assigned to different dendritic locations. The solid line represents a quadratic fit to the data. The dashed line is the corresponding quadratic fit from *Figure 4E*. (**D**) Peak voltage-gated calcium (left panel, dots) and plasticity threshold dynamics (lines), in dendrite 1, over training epochs. The upper threshold ($\theta_{inh,high}$) moves toward peak calcium levels while the lower threshold ($\theta_{inh,low}$) moves toward the next highest level. (Right) Inhibitory synaptic conductances for the synapses in dendrite 1. Strengthened inhibitory synapses prevent excitation by the corresponding excitatory features (here 'red' and 'strawberry'), while weakened inhibitory synapses allow excitation by their corresponding features (here 'yellow' and 'banana'). (**E**) Excitatory synaptic conductances (top) and calcium levels and kernel dynamics (below) over learning. The specific conductances in the top panel corresponding to the kernel dynamics and calcium levels in the bottom panels are indicated with arrows. Note that kernel and calcium dynamics for the example 'yellow' synapse refer to the only 'yellow' synapse in the top panel which initially

*Figure 5 continued on next page*

*Figure 5 continued*

weakens and is later strengthened. The solid line shows the midpoint of the long-term potentiation (LTP) kernels. *Max* refers to the peak NMDA calcium during a single stimulus. The darker shaded regions represent the LTP kernel, and the lighter shaded ones show the wider metaplasticity kernel.

The online version of this article includes the following figure supplement(s) for figure 5:

**Figure supplement 1.** Learning with and without inhibitory plasticity.

how inhibition suppresses calcium activity arising from irrelevant stimuli while allowing that evoked by relevant stimuli (*Figure 5D*, left panel). The high threshold level of calcium for inhibitory synapses ($\theta_{\text{inh, high}}$) followed the highest excitatory synaptic activity, which in this case corresponded to the 'yellow banana' input.

We also compared the performance for different feature configurations with added inhibitory synapses to the results for the excitatory-only setup from *Figure 4B*. The results show not only that learning with inhibitory plasticity is faster, i.e., requires fewer training examples, but also achieves high performance, nearing 100%. In the configurations with all four features in one dendrite, 7/13 learned the NFBP, compared to 5/13 without inhibition (shown in the bar plot inset in *Figure 5B*).

To understand why inhibitory plasticity improved learning, we examined the dynamics of L-type calcium during early learning. The presence of inhibition resulted in slightly lower initial L-type calcium levels compared to the excitatory-only setup, leading to reduced LTD at the start. This reduction in LTD helped prevent excessive weakening of synapses, increasing the likelihood of selecting one combination over others (*Figure 5—figure supplement 1B*, showing excitatory synaptic weights without (left) and with inhibitory plasticity (right): in the setup without inhibitory plasticity, all four features weaken over time due to competition, while in contrast, the setup including inhibitory plasticity demonstrates better-regulated learning and more stable feature selection).

Inhibitory plasticity further enhanced performance when synaptic cluster locations were varied, with improvements evident across nearly all tested dendritic pairs and especially pronounced at distal sites (*Figure 5C*; cf. the orange and blue markers).

Without inhibition, it was difficult for synapses on distal dendrites to differentiate between relevant and irrelevant stimuli. In the middle of the learning, setups without inhibition typically spiked too much compared to setups including inhibition (e.g. spike for 'red banana' in the left panels of *Figure 5—figure supplement 1A*). This, in turn, caused increased LTD and decreased weights in the feature-unspecific synapses, leading to too little spiking at the end of the learning (e.g. no spike for 'yellow banana' in the right panels of *Figure 5—figure supplement 1A*). With inhibition, on the other hand, the contrast between relevant and irrelevant stimuli was larger. Larger contrast, in turn, gave less spiking for the wrong combination in the middle of learning and thereby less reduction of the weights of the feature-unspecific synapses. Therefore, large excitation in a single dendrite, combined with a larger drive of feature-unspecific synapses, was enough to cause spiking in relevant stimuli at the end of learning, while spiking in irrelevant stimuli was suppressed (*Figure 5—figure supplement 1A*, red traces).

In summary, incorporating inhibitory plasticity suggests that inhibitory synapses may fine-tune dendritic responsiveness and enhance NFBP performance by preventing excitation from irrelevant stimuli, which in turn allows for more rapid, robust, and accurate learning.

## Inhibitory plasticity improves performance with randomly distributed inputs

We finally challenged our plasticity rule by relaxing the assumption that single features are represented by pre-clustered synapses on specific dendritic branches. The behavior of the synaptic plasticity rule was therefore investigated using a setup where 200 excitatory synapses, representing the different features, were randomly distributed across 30 dendrites. In this setup, each feature was represented by 40 synapses and an additional 40 represented the feature-unspecific synapses. *Figure 6A* illustrates the setup and exemplifies the pre- and post-learning synaptic weights for both excitatory and inhibitory synapses. Our objective was to examine the learning dynamics in the absence of assumed synaptic clustering and to determine the capability of the single neuron to learn the NFBP. To address the reduced effectiveness of non-clustered synaptic inputs in eliciting sufficient depolarization and calcium influx – key for learning and synaptic plasticity – we slightly increased the initial weights

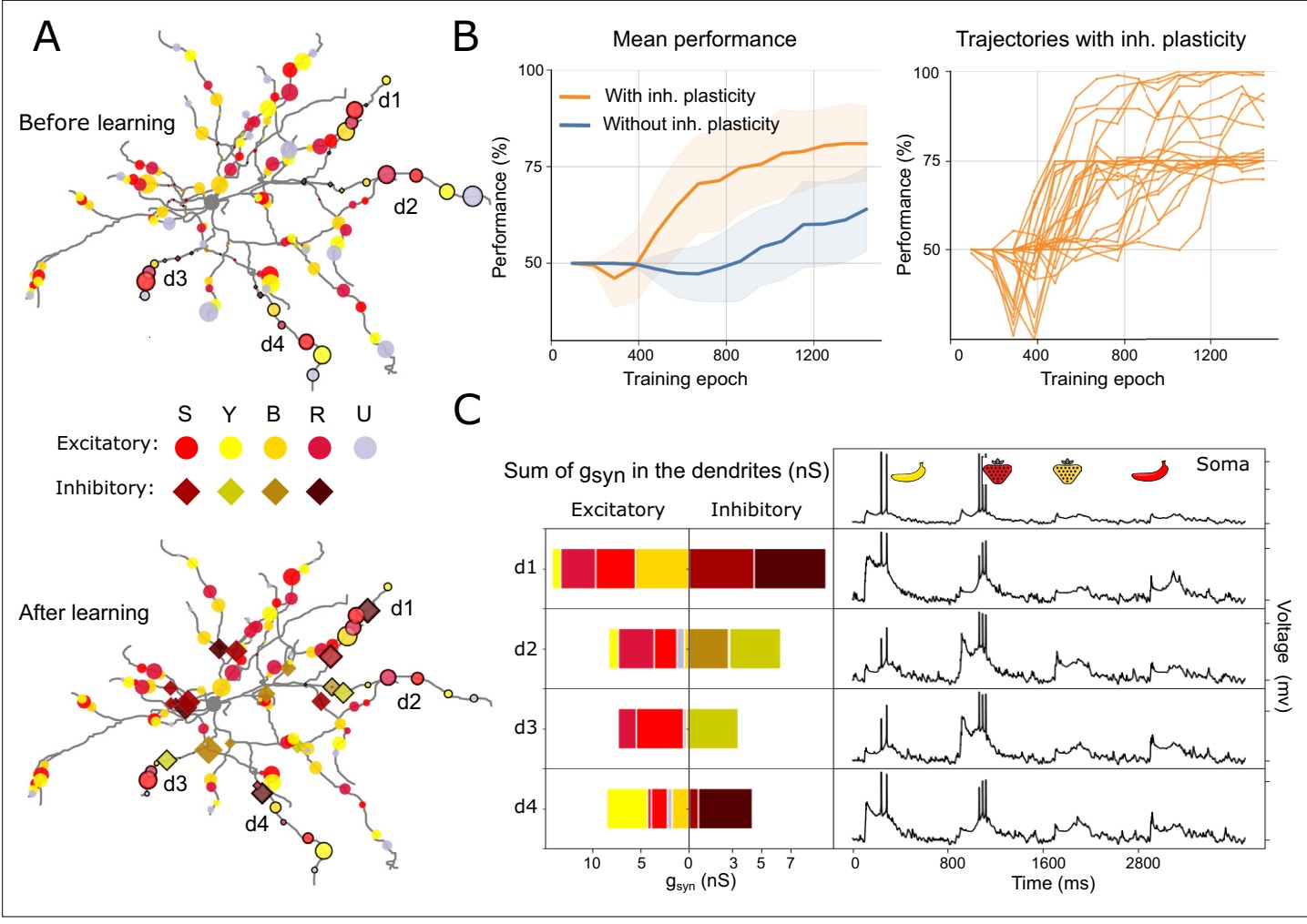

**Figure 6.** Performance analysis of learning using distributed synaptic inputs. (**A**) Example illustration of synaptic distribution before (top) and after learning (bottom) of the 200 excitatory and 60 inhibitory inputs. S – strawberry, Y – yellow, B – banana, R – red, U – feature-unspecific inputs. Bordered markers (circles and diamonds) indicate synapses whose conductances are shown in (**C**). (**B**) Performance over training epochs with and without inhibitory plasticity. (Left) Mean performance of the setups with and without inhibitory plasticity. Shaded areas show standard deviation. (Right) Individual traces for the setup with inhibitory plasticity. Each of these individual trials (n = 31 in total) uses a unique random distribution of synapses. (**C**) Example of summed synaptic conductances (left) and voltage (right) in the soma, and four example dendrites (d1–d4) of the synaptic distribution in (**A**) (corresponding to a trial where the nonlinear feature binding problem (NFBP) is successfully solved). The sums of both excitatory (left) and inhibitory (right) inputs are shown.

The online version of this article includes the following figure supplement(s) for figure 6:

**Figure supplement 1.** Other learning paradigms.

compared to the clustered setup to 0.3 ± 0.1 (around 0.75 nS). A branch-specific mechanism for creating nonlinearities was also introduced by assuming that glutamate spillover occurs in all co-activated synapses on the same dendritic branch. This hypothetical construct assumes that concurrent activation of a critical mass of co-activated synapses in a single dendritic branch would trigger both plateau potentials and a rise in calcium levels, endowing the dendrites with supralinear responses even without closely clustered synapses.

Starting with an excitation-only setup resulted in a moderate mean performance of about 65% (*Figure 6*, 'Without inhibitory plasticity'). However, on a linear task (e.g. learning only 'yellow banana' or 'red strawberry') with randomly placed synapses, the performance was close to 100% (even in the so-called subthreshold learning, starting from an initial condition without spiking; see *Figure 6— figure supplement 1B and C*).

We next extended our investigation by including 60 plastic inhibitory synapses, 15 for each feature, dispersed randomly over the 30 dendrites. Performance improved with the introduction of these additional nonlinearities, especially in combination with the branch-specific spillover mechanisms. In the right panel of *Figure 6B*, we show that, with inhibition, most individual distributions (22/31) reach a performance of about 75%, while a subset of distributions (5/31) achieve performance close to 100%. There are also a few distributions that partially solve the NFBP, reaching a performance between 75% and 100% (4/31).

To explore the model's ability to handle increased feature complexity, we extended the task and explored two variants: one with 9 feature combinations, including one nonlinear combination, and another, more challenging variant with 25 feature combinations, including two nonlinear combinations (see *Figure 6—figure supplement 1A*). In these extended simulations, the model achieved accuracies of 80% and 75% for the 9- and 25-feature tasks, respectively, as shown in *Figure 6—figure supplement 1A*.

For a more granular analysis of the successful cases with randomly distributed synapses and assumed branch-specific plasticity, we looked in detail at the conductances of an example that successfully learned the NFBP. In *Figure 6C*, we present the somatic and dendritic voltage responses for four dendritic branches (d1–d4) where relevant stimuli were encoded. The voltage traces display somatic and dendritic responses to all four stimuli after learning. Additionally, the cumulative synaptic conductances for both excitatory and inhibitory inputs were calculated at the midpoint of each dendritic branch. Notably, dendrites 1 and 4 show enhanced excitatory inputs for the 'yellow banana' pattern and increased inhibitory inputs for the 'red strawberry' pattern, while dendrites 2 and 3 exhibit the reverse arrangement. The synapses in dendrites 1 to 4, whose weights are shown in panel C, are marked in *Figure 6A* with black borders around the markers.

These results show that for an adequate random innervation of distributed synapses – where the necessary features for a relevant stimulus innervate the same dendritic branch with enough synapses – that stimulus can be stored on that dendrite. In this way, the NFBP can also be learned if the two relevant stimuli are encoded on different dendrites and each of them can trigger a supralinear dendritic response. Since, on average, the two features representing a relevant stimulus do not innervate a single dendrite with enough synapses, the stimulus is not stored in a single dendrite but is distributed across the dendritic tree. And since distributed synapses summate more linearly at the soma, only one of the relevant stimuli can typically be encoded by the neuron.

## Discussion

In this article, we studied whether single neurons can solve linearly non-separable computational tasks, represented by the NFBP, by using a biophysically detailed multicompartment dSPN model. Based on the synaptic machinery of corticostriatal synapses onto dSPNs, we propose a learning rule that uses local synaptic calcium concentration and dopamine feedback signals: rewards for relevant stimuli and omitted rewards for irrelevant stimuli. Assuming first that single features in the NFBP are represented by clustered synapses, we show that the learning rule can solve the NFBP by strengthening (or stabilizing) synaptic clusters for relevant stimuli and weakening clusters for the irrelevant stimuli. The feature combinations for the relevant stimuli, stored in strengthened synaptic clusters, trigger supralinear dendritic responses in the form of plateau potentials, which is an important ingredient for the solution of the NFBP, as plateaus significantly increase the likelihood of neuronal spiking in SPNs in a robust way (*Du et al., 2017*).

The location of the synaptic clusters along the dendrites influenced the performance on the NFBP. In our model, the region for optimal performance was around 150 micrometers away from the soma, at about the same distance as where the somatic depolarization and induced spike probability, following activation of clustered synaptic input, was largest in the model by *Lindroos and Hellgren Kotaleski, 2021*. Clusters placed further away produce smaller somatic depolarizations, due to dendritic filtering (*Major et al., 2008*), and as a consequence, do not control the likelihood of somatic spiking as decisively. As the supralinear dendritic response is necessary for discriminating the relevant from the irrelevant stimuli, the performance with distally placed clusters decreases somewhat.

Further, we relaxed the assumption of pre-clustered synapses by using randomly distributed synapses for each feature. In this scenario, only one relevant stimulus is sometimes learned in a dendritic branch by the randomly distributed synapses. In the random setup, the glutamate spillover

models were updated to include the synapses in the whole dendritic branch, building on the notion that the single branch acts as a single computational unit (*Branco and Häusser, 2010*; *Losonczy and Magee, 2006*). The realism of this spillover assumption remains open to debate. In actual dendritic branches, however, the diffusion of signaling molecules likely plays a crucial role in both synaptic plasticity at existing synapses and for locally induced structural plasticity (*Nishiyama and Yasuda, 2015*; *Chater et al., 2024*). These processes may both enhance branch-specific supralinearities in qualitatively similar ways as when using spillover, but the mechanisms are not explored in the current model.

By using a phenomenological inhibitory plasticity rule based on the Bienenstock-Cooper-Munro (BCM) formalism (*Bienenstock et al., 1982*), we also show that inhibitory synapses can significantly improve performance on the NFBP. This is in line with earlier theoretical studies where negative synaptic weights were required to solve the NFBP (*Schiess et al., 2016*). In our setup with pre-existing synaptic clusters, inhibitory synapses made learning faster and increased performance by inhibiting supralinear NMDA responses for the irrelevant stimuli. This was specifically true in distal dendrites where the input impedance is higher (*Branco et al., 2010*). The threshold for plateau potential initiation is also lower in distal dendrites compared to proximal (*Losonczy and Magee, 2006*), which likely will further extend the influence of inhibition in this region (*Doron et al., 2017*; *Du et al., 2017*). Similarly, in the scenario with distributed synapses, inhibition enables one of the relevant stimuli to be reliably encoded in the dendritic branch by strengthening the inhibitory synapses for features different from those of the encoded stimulus. Together, it therefore seems like inhibition not only has a role in learning (*Chen et al., 2015*; *Cichon and Gan, 2015*), but also improves the ability of the neuron to discriminate between stimuli with shared features.

In calcium-dependent plasticity models, a 'no man's land' has sometimes been proposed as a region where synaptic weights remain largely unchanged within a specific calcium range, reducing sensitivity to fluctuations and improving stability (*Lisman, 2001*; *Moldwin et al., 2024*). For our inhibitory rule, which builds on a BCM-like mechanism, introduction of such a mechanism in future work could make the system less sensitive to fluctuations in calcium concentration.

Although our learning rule infrequently solves the NFBP when used with only randomly distributed synapses, it can always learn to perform a linearly separable task, such as learning to respond to only one relevant stimulus (e.g. red strawberry). Moreover, the learning rule is general enough so that in addition to the feature-specific inputs related to the task, it can handle feature-unspecific inputs that might or might not be related to the NFBP. Finally, the learning rule is always 'on', continuously updating synapses with each stimulus presentation, which is a more realistic mechanism compared to using separate training and testing phases as in the field of machine learning. The synaptic weights automatically stabilize in the model when the performance improves, with metaplasticity playing a crucial role in this process. That is, rewards are then seen very regularly as the neuron has learned to spike for the relevant stimuli, while omitted rewards rarely occur as the neuron stays silent when the irrelevant stimuli are provided.

When formulating the learning rule in this article, our goal was to base it on what is known regarding the synaptic machinery in corticostriatal synapses. This implied that the learning rule is based on the local calcium activity and on dopamine signals. Feedback from the dopamine system can also be viewed as an innate, evolutionarily encoded 'supervisor', which instructs neurons which feature combinations are beneficial and which ones should be avoided. However, in our case, we do not use additional excitation to promote somatic spiking for only the relevant feature combinations, and in that sense, the learning rule does not require a supervised learning paradigm. However, since the SPNs rest at very hyperpolarized membrane potential, our setup includes distributed excitatory inputs which are feature-unspecific in order to make sure that the neuron spikes for all stimuli, especially at the beginning of training. These additional inputs are on average weakened as learning progresses (as they are activated for all stimuli and thus often receive negative feedback). That general or noisy inputs are reduced during learning is in line with the observed reduction of execution variability during motor learning as a novice becomes an expert (*Kawai et al., 2015*). Specifically, the underlying neuronal representation of corticostriatal synapses undergoes a similar change during learning (*Santos et al., 2015*).

Since each synapse has its own calcium response, it is important for the learning rule to be able to follow individual synaptic activities. The LTP plasticity kernel in our model is for this reason itself plastic (metaplasticity), meaning that it changes its calcium dependence as a function of the reward history

(dopamine peaks and pauses). This setup helps the model separate clustered synaptic input from the feature-unspecific input at the beginning of training, as only clustered synapses will see enough calcium to fall within the LTP plastic range (LTP kernel).

We further use an asymmetric metaplasticity rule, where negative feedback causes a larger shift of the LTP kernel than a positive. This was necessary in order to prevent LTP in synapses that should ultimately undergo LTD. Hence, similarly to the classical loss-aversion tendency described in economic decision theory (*Kahneman and Tversky, 1979*), the model, through its requirement for asymmetric metaplasticity, predicts that negative feedback has a bigger impact on changing a well-learned behavior at the single-cell level than positive feedback. Dopamine signaling has also been linked as a neural substrate to the decision-making theory mentioned above (*Stauffer et al., 2016*).

It is not known whether single neurons solve the NFBP or other linearly non-separable tasks. However, many brain nuclei receive convergent inputs from numerous other brain nuclei, acting as integratory hubs (*van den Heuvel and Sporns, 2013*). Since feature binding evidently occurs in the brain, and functional clusters for single features such as visual stimulus orientation, receptive fields, color, or sound intensity exist on single neurons (*Chen et al., 2011*; *Wilson et al., 2016*; *Iacaruso et al., 2017*; *Scholl et al., 2017*; *Ju et al., 2020*), it is possible that the NFBP is a relevant task for neurons to solve. How brain regions with synaptic machinery different from that of striatal dSPN might solve the NFBP remains an open question, and reliance on other neuromodulatory signals may be part of the answer. For example, in the striatum, the indirect pathway SPNs (iSPN) have analogous synaptic machinery to the one in dSPNs, requiring calcium influx from the same sources for LTP and LTD, but are differently responsive to dopamine (*Shen et al., 2008*). In iSPNs, a dopamine pause, together with a peak in adenosine, is required to trigger LTP, whereas a dopamine peak without peaks in adenosine rather promotes LTD (*Shen et al., 2008*; *Nair et al., 2015*). Therefore, we expect that an analogously formulated learning rule will also solve the NFBP in iSPNs, activating them for irrelevant feature combinations to, e.g., suppress movement, and suppressing their activity for relevant feature combinations to facilitate movement. In addition, LTP in dSPNs is significantly facilitated by co-regulation of other neuromodulatory systems, such as when there is a coincident acetylcholine pause with the dopamine peak, which we have not explicitly included in the model (*Nair et al., 2015*; *Bruce et al., 2019*; *Reynolds et al., 2022*).

## Materials and methods

In this paper, we introduce a local, calcium- and reward-based synaptic learning rule, constrained by experimental findings, to investigate learning in SPNs. The learning rule operates based on the changes in local calcium concentration resulting from synaptic activation patterns that evoke dendritic nonlinearities, such as plateau potentials. Such events affecting the postsynaptic calcium concentration can enable learning of the NFBP. The learning rule is embedded in a biophysically detailed model of a dSPN built and simulated in the NEURON software, v8.2 (*Carnevale and Hines, 2006*). Here, we will focus on the setup of the learning rule and only give a short summary of the neuron and synapse models and emphasize changes compared to previously published versions. For a detailed description of the neuron model setup, see *Lindroos et al., 2018*; *Lindroos and Hellgren Kotaleski, 2021*. The modeling of plateau potentials is explained in *Trpevski et al., 2023*. See also the section *Code and data availability* below.

### Neuron model

In short, the dSPN model used here was taken from the library of biophysically detailed models in *Lindroos and Hellgren Kotaleski, 2021*, including a reconstructed morphology and all of the most influential ion channels, including six calcium channels, each with its own voltage dependence and dendritic distribution. In accordance with *Trpevski et al., 2023*, the model was further extended with synaptic spines on selected dendrites. Each spine was modeled as two additional compartments consisting of a neck and a head region and contained voltage-gated calcium channels of types R ($Ca_v2.3$), T ($Ca_v3.2$ and $Ca_v3.3$), and L ($Ca_v1.2$ and $Ca_v1.3$); the addition of explicit spines did not change the basic behavior of the model, such as the response to current injections, etc.

## Calcium sources used in learning

The intracellular calcium concentration is separated into distinct pools that are used during the learning process. For learning in glutamatergic synapses, one pool for NMDA-evoked calcium concentration ($[Ca]_{\mathrm{NMDA}}$) is used, and another for L-type calcium concentration ($[Ca]_{\mathrm{L-type}}$), to reflect the different synaptic plasticity responses of the dSPN's biochemical machinery to these two calcium sources in the corticostriatal synapse (*Shen et al., 2008*; *Fino et al., 2010*; *Plotkin et al., 2013*). Both pools are based on the calcium influx from the corresponding source (NMDA and the L-type channels $Ca_v1.2$ and $Ca_v1.3$, respectively). Inspired by findings that calcium from various voltage-dependent calcium channels affects GABAergic plasticity in different systems (*Kurotani et al., 2008*; *Hulme and Connelly, 2014*; *Udakis et al., 2020*), we also incorporate a third calcium pool based on calcium influx from all voltage-gated channels (T-, R-, L- and N-type, $[Ca]_v$). All pools include extrusion mechanisms in the form of a calcium pump, as well as a one-dimensional time decay. The calcium pump follows the implementation in *Wolf et al., 2005*. The parameters for the $[Ca]_{\mathrm{NMDA}}$ model were manually tuned to match the [Ca] amplitudes and durations reported in *Dorman et al., 2018*. The voltage-gated calcium channel conductances in the spines were also manually tuned to match the relative calcium proportions in *Carter and Sabatini, 2004*; *Higley and Sabatini, 2010*, as well as the calcium amplitudes due to stimulation with backpropagating action potentials (*Shindou et al., 2011*). Spatial calcium diffusion was not included in the present model.

## Glutamatergic synaptic input

Clustered synapses were modeled following *Trpevski et al., 2023*, using a version of the saturating synapse model in *Destexhe et al., 1994*. The synaptic model included AMPA and NMDA conductances activated on spines, as well as extrasynaptic NMDA conductances located on the dendritic shafts adjacent to the spine. The strength of the individual synapses on spines was scaled using a weight parameter ($w$) taking on positive values. More explicitly, in our model, the parameter $w$ represents the percentage of the maximum synaptic conductance $g_{\mathrm{max}}$. For instance, when $w = 0.5$, the synaptic conductance ($g$) is $0.5 \cdot g_{\mathrm{max}}$. Importantly, changes in $w$ affect both NMDA and AMPA receptor conductances simultaneously, as both share the same scaling factor. Extrasynaptic conductances were activated following sufficiently large stimulation of nearby spines, resulting in glutamate spillover. Extrasynaptic NMDA synapses were included because they provide robust dendritic plateaus (*Trpevski et al., 2023*). The maximal conductance of both synaptic and extrasynaptic NMDARs was set to 2.5 nS, which is lower than the 3.5 nS used in *Trpevski et al., 2023*. This reduction in conductance was implemented to compensate for the increased number of inputs in the model. In this model, spillover was triggered when the sum of the weights of active synapses in a cluster exceeded a threshold value of 2. For instance, eight synapses with individual weights greater than 0.25 would activate spillover. This threshold was lower than the value of 4 used in *Trpevski et al., 2023*, in order to facilitate spillover and enhance nonlinearity in the context of distributed synapses. However, this scaling only affects the size of the clustered inputs presumed to give rise to plateau potentials, which likely varies across different neuronal subtypes or even between distinct dendritic branches within the same subtype. In the NEURON simulation environment, an integrate-and-fire cell was used to generate a spike in extrasynaptic NMDARs when the spillover threshold is reached. As synaptic weight increases, fewer synapses are required to meet the threshold. This also links LTP to enhanced glutamate spillover. In addition to task-related inputs, the neuron also receives background noise, modeled as excitatory and inhibitory synapses with fixed weights distributed across the dendrites. Since the number of electrical compartments in the model is significantly lower than the number of synapses in a real SPN, these converging synapses were represented by a single non-saturating synapse with input frequency scaled accordingly. This was implemented using a dual exponential synapse model based on the NEURON `exp2syn` mechanism following *Lindroos and Hellgren Kotaleski, 2021*, with time constants and other parameters adapted from *Hjorth et al., 2020*.

## Learning rule

### Excitatory synaptic plasticity

The learning rule is grounded in experimental evidence, indicating that striatal LTP relies on NMDA receptor activation and the presence of dopamine, whereas LTD depends on the activation of L-type calcium channels ($Ca_v1.3$) and mGluR receptors in the absence of dopamine (*Shen et al., 2008*; *Fino*

*et al., 2010*; *Plotkin et al., 2013*; *Yagishita et al., 2014*; *Fisher et al., 2017*; *Shindou et al., 2019*). Under basal dopamine levels, no significant synaptic plasticity is assumed to occur. This rule outlines a reward-based learning mechanism where a dopamine peak enhances synaptic weight via an LTP plasticity kernel, while a dopamine pause reduces synaptic weight through an LTD kernel. Thus, dopamine functions as a switch, determining whether LTP or LTD pathways are activated (as illustrated in *Figure 1D*). Notably, synaptic weights affect both AMPA and NMDA receptors, consistent with findings that the NMDA-to-AMPA ratio remains unchanged following LTP (*Watt et al., 2004*). Additionally, the previously described spillover mechanism is linked to synaptic weights, representing an attempt to associate plasticity with increased glutamate spillover resulting from the retraction of astrocytic processes from the spine (*Henneberger et al., 2020*). The excitatory synaptic plasticity rule encompasses key components such as LTP, LTD, and metaplasticity, each of which will be elaborated upon in the subsequent sections.

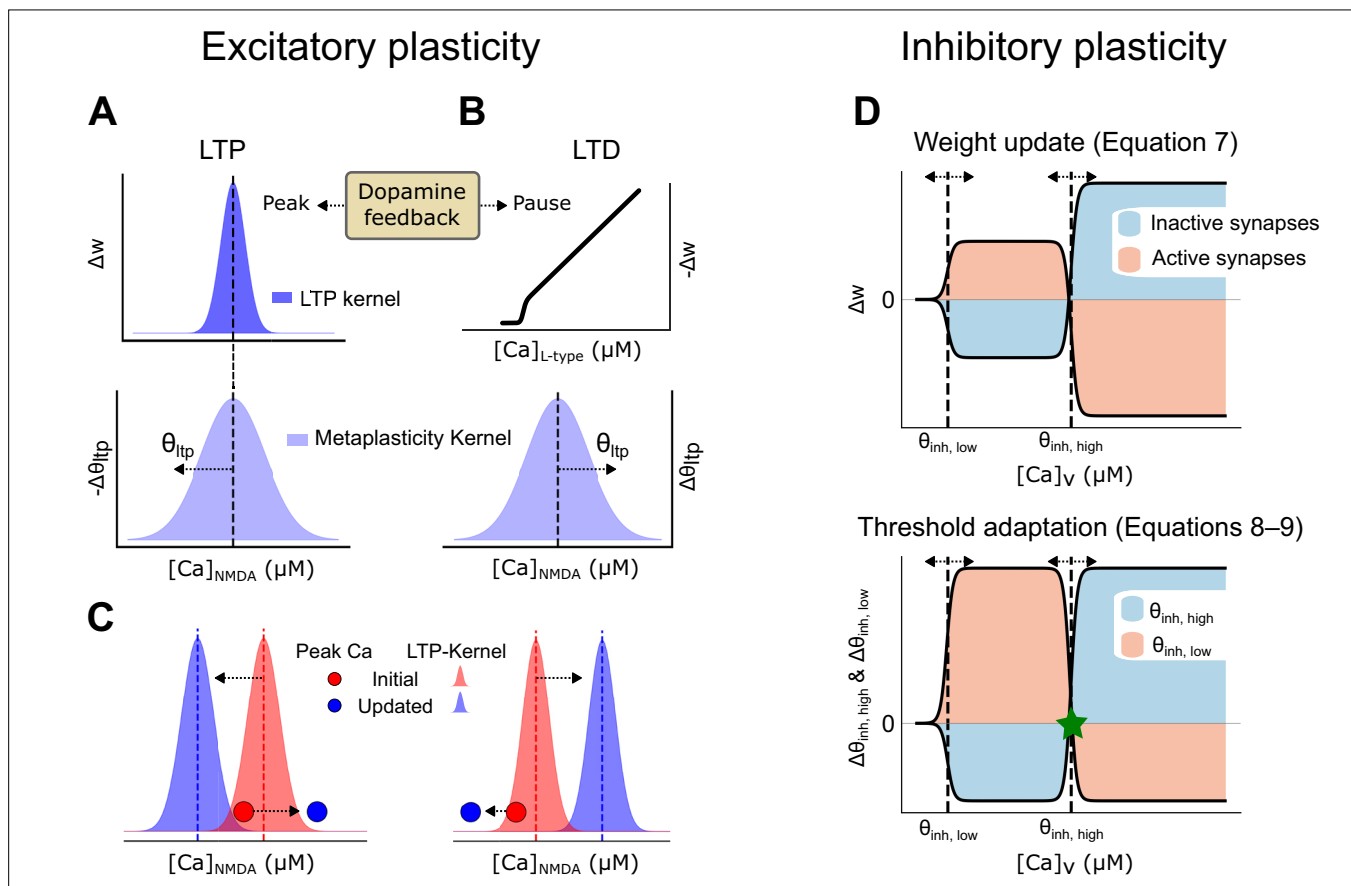

**Figure 7.** Synaptic plasticity rules: calcium and dopamine interactions in synaptic weight modification. (**A**) Synaptic weight updates following a dopamine peak. (Top) The long-term potentiation (LTP) kernel is a bell-shaped curve describing the amount of weight increase, which happens over a region of $[Ca]_{NMDA}$. (Bottom) A wider bell-shaped kernel, i.e., the metaplasticity kernel, determines how the LTP kernel is shifted along the calcium level ($[Ca]_{NMDA}$) axis following a peak in dopamine. (**B**) Synaptic weight updates following a dopamine pause. (Top) The long-term depression (LTD) plasticity kernel. The LTD threshold is constant and set at 70 nM. (Bottom) Metaplasticity describing how the LTP kernel shifts along the calcium axis following a dopamine pause. (**C**) A schematic of how the LTP kernel is shifted following a dopamine peak (left) and pause (right), together with examples of NMDA calcium levels before and after the shift. The NMDA calcium levels change following the potentiation or suppression of the synapse (illustrated with the red circle jumping to the blue circle). The LTP kernel also moves as indicated by the arrows and colors following a peak or a pause, respectively. (**D**) Illustration of the inhibitory plasticity rule. (Top) Changes in synaptic weight for active (beige) and inactive (blue) synapses based on voltage-dependent calcium levels in the dendritic shaft at the location of the inhibitory synapse. The dashed lines show the minimum/lower ($\theta_{inh,low}$) and maximum/higher ($\theta_{inh,high}$) thresholds. (Bottom) Functions for updating the thresholds of the upper panel, depending on the voltage-dependent calcium level. The asterisk denotes the calcium level where the curves for active and inactive synapses meet, which is a point of zero update.

## LTP

The LTP process is initiated by elevated dopamine levels, following calcium influx through NMDA receptors. Synaptic strength is adjusted using the LTP kernel, a bell-shaped function where the maximum increase occurs when peak NMDA calcium levels are at the kernel's midpoint; deviations above or below this midpoint result in smaller increases (*Figure 7A*, top panel). The left side of the LTP kernel represents the lower calcium threshold necessary for LTP induction, while the right side indicates an upper calcium threshold beyond which LTP does not occur. This framework, in conjunction with metaplasticity, establishes an upper boundary for synaptic strength, ensuring that synaptic weights do not exceed a certain limit (*Zenke and Gerstner, 2017*; *Zenke et al., 2017*). Since each synapse experiences unique calcium levels (due to different synaptic conductance, local input resistance, etc.), adjusting the calcium dependence of the LTP kernel allows for individualized tuning of synaptic weights (see *Figure 7A and C* for illustrations).

The update of the synaptic weights follows a rule formulated using the derivative of the sigmoid function (*Equation 2*). The sigmoid function (*Equation 1*) transforms the calcium concentration into a normalized range between 0 and 1:

$$\sigma(Ca, \theta, \beta) = \frac{1}{1 + e^{-\beta(Ca - \theta)}} \tag{1}$$

where $\vartheta$ represents the midpoint of the transition, representing the calcium concentration at which the function increases most rapidly, and $\beta$ determines the steepness of this transition.

The derivative of the sigmoid function, given by:

$$\sigma'(Ca, \theta, \beta) = \beta \cdot \sigma(Ca, \theta, \beta) \cdot \left(1 - \sigma(Ca, \theta, \beta)\right) \tag{2}$$

produces a bell-shaped curve, which forms the LTP kernel. This kernel peaks at the midpoint $\vartheta$, where synaptic modification is maximized and decays symmetrically on each side. The parameter $\beta$ here controls the width of the kernel, thereby regulating the sensitivity of synaptic weight changes to calcium fluctuations.

Based on this general description of the mathematical formulas underlying the LTP kernel, the LTP learning rule for the synaptic weights is formulated as:

$$\Delta w_{\text{ltp}} = \eta_{\text{ltp}} \cdot \sigma'(Ca_{\text{NMDA}}, \theta_{\text{ltp}}, \beta_{\text{ltp}}) \tag{3}$$

where $\Delta w_{\text{ltp}}$ represents the change in synaptic weight, $\eta_{\text{ltp}}$ is the learning rate, and $[Ca]_{\text{NMDA}}$ denotes the calcium concentration mediated by NMDA receptors. The parameters $\theta_{\text{ltp}}$ and $\beta_{\text{ltp}}$ define the midpoint and slope of the sigmoid curve specific to LTP. The parameter values used in these equations are provided in *Table 1*.

**Table 1.** Excitatory plasticity parameters.

| Parameter | Description | Value |
|---|---|---|
| $\eta_{\text{ltp}}$ | Learning rate for $[Ca]_{\text{NMDA}}$-dependent plasticity | $1.5 \times 10^{-5}$ µS mM$^{-1}$ ms$^{-1}$ |
| $\theta_{\text{ltp}}$ | Midpoint of LTP kernel (initial value) | 0.02 mM |
| $\beta_{\text{ltp}}$ | Controls width of LTP kernel | $1.0 \times 10^{3}$ mM$^{-1}$ |
| $\eta_{\text{ltd}}$ | Learning rate for $[Ca]_{\text{L-type}}$-dependent plasticity | $3 \times 10^{-3}$ ms$^{-1}$ mM$^{-1}$ |
| $\theta_{\text{ltd}}$ | Constant LTD threshold | $7 \times 10^{-5}$ mM |
| $\beta_{\text{ltd}}$ | Controls steepness of LTD threshold | $1.0 \times 10^{5}$ mM$^{-1}$ |
| $\eta_{\text{s,ltp}}$ | Rate for shifting $\theta_{\text{ltp}}$ during dopamine peaks | $1 \times 10^{-7}$ mM$^{2}$ ms$^{-1}$ |
| $\eta_{\text{s,ltd}}$ | Rate for shifting $\theta_{\text{ltp}}$ during dopamine pauses | $4 \times 10^{-7}$ mM$^{2}$ ms$^{-1}$ |
| $\beta_{\text{mp}}$ | Controls width of metaplasticity kernel | 334 mM$^{-1}$ |

## LTD

The LTD process is triggered by a dopamine pause and is dependent on L-type calcium. The LTD plasticity kernel (*Equation 4*; see *Figure 7B*, *top* panel) describes a threshold calcium level necessary for LTD to occur (*Shindou et al., 2011*). Once this threshold is exceeded, the decrease in synaptic weight is linearly proportional to the amplitude of the peak calcium level and is scaled by the learning rate ($\eta_\text{ltd}$). The peak calcium threshold is implemented with a sigmoid function whose slope parameter ($\beta_\text{ltd}$) is set high so the curve approximates a step function. The threshold is fixed at 70 nM. The NMDA calcium dependence of the LTP kernel is also increased during LTD, as described in the next section (*Figure 7B*, *bottom* panel, and *Figure 7C*, right panel).

$$\Delta w_\text{ltd} = \eta_\text{ltd} \cdot Ca_\text{v1.3} \cdot \sigma(Ca_\text{v1.3}, \theta_\text{ltd}, \beta_\text{ltd}) \cdot w_\text{ltd} \tag{4}$$

## Metaplasticity

Metaplasticity is a form of regulatory mechanism changing the state of the synapses in such a way as to influence subsequent learning (*Abraham, 2008*). In our model, we implemented this as a reward-dependent change of the calcium concentration over which the LTP kernel was operating. Specifically, activation of the LTD pathway triggered a shift of the LTP kernel toward higher calcium concentrations while activation of the LTP pathway pushed it in the opposite direction.

Metaplasticity was also implemented using a bell-shaped kernel with the same midpoint as the LTP kernel, but with a wider calcium dependence. The position of the metaplasticity kernel is updated at the same time with the LTP kernel, so that both kernels remain centered at the same calcium level (*Figure 7A–C*). This setup with the two moving kernels together allowed for a wide range of calcium levels to induce plasticity.

In addition, using a wider metaplasticity kernel ensured the following:

- Following activation of the LTP pathway (i.e. after a dopamine peak, see *Figure 7C*, left panel for an illustration):
  - In synapses with low calcium levels, the kernel will be shifted closer to the observed calcium level and thereby eventually enable LTP in synapses that are regularly activated during rewards, even though they initially don't generate big elevations in calcium.
  - In synapses with already high calcium levels (above the kernel midpoint), the kernel will be shifted away from the observed calcium level, and thereby protect the synapse from excessive LTP and instead stabilize the weight (as illustrated with the red circle moving to the blue one in *Figure 7C*, left panel).
- Following activation of the LTD pathway (i.e. after a dopamine pause), the kernel will be shifted toward higher calcium levels and thereby reduce the likelihood of LTP in these synapses that are weakened following LTD (see *Figure 7C*, right panel for an illustration).

The update of the LTP kernel was further asymmetric following activation of the LTD or LTP pathways in such a way that the LTD pathway caused a larger shift of the metaplasticity kernel than the LTP pathway. This further reduced the likelihood of inducing LTP in synapses often participating in LTD or in synapses randomly activated with regard to the dopamine feedback signal.

It also allows initially weaker synapses to be recruited for LTP if they are more consistently co-active with a reward than with an omitted reward (an illustration of this can be seen in *Figure 5E*, where one of the yellow synapses is strengthened in the middle of the training session).

The LTP kernel was updated according to *Equation 5*, based on the learning rate ($\eta_s$), the peak NMDA calcium level ([Ca]$_\text{NMDA}$), and the slope of the sigmoid curve ($\beta_\text{mp}$). The value of the slope $\beta_\text{mp}$ creates a wider bell-shaped function in *Equation 5*. The rate $\eta_s$ captures the described asymmetry in how the metaplasticity kernel is shifted in response to dopamine feedback, where the specific rates following dopamine peaks and pauses are denoted $\eta_{s,\text{ltp}}$, and $\eta_{s,\text{ltd}}$, respectively. Here, $\eta_{s,\text{ltd}}$ is four times larger than $\eta_{s,\text{ltp}}$ (see *Table 1*).

$$\Delta\theta_\text{ltp} = \eta_s \cdot \sigma'(Ca_\text{NMDA}, \theta_\text{ltp}, \beta_\text{mp}) \tag{5}$$

## Pseudocode for the excitatory plasticity rule

**Inputs:**

- [Ca]$_\text{NMDA}$ : Calcium level associated with the NMDA channel

- $[Ca]_{\text{L-type}}$ : Calcium level associated with the L-type channel
- Dopamine level : Dopamine level (1 for peak, −1 for pause, 0 for basal)

**Output:**

- $\Delta w$: Change in synaptic weight

---

Algorithm 1:

---

- `begin`
  - `if dopamine_level == 1 then`
  - `# Compute LTP weight change and adjust kernel midpoint for metaplasticity`
    - `*` $\Delta w$ = solve **Equation 3** with $[Ca]_{\text{NMDA}}$, $\eta_{\text{ltp}}$, $\theta_{\text{ltp}}$, and $\beta_{\text{ltp}}$
    - `*` update $\theta_{\text{ltp}}$ using **Equation 5** with $[Ca]_{\text{NMDA}}$, $\eta_{s,\text{ltp}}$, $\beta_{\text{mp}}$
  - `else if dopamine_level == -1 then`
  - `# Compute LTD weight change and adjust LTP kernel midpoint for metaplasticity`
    - `*` $\Delta w$ = solve **Equation 4** with $[Ca]_{\text{L-type}}$, $\eta_{\text{ltd}}$, $\theta_{\text{ltd}}$, and $\beta_{\text{ltd}}$
    - `*` update $\theta_{\text{ltp}}$ using **Equation 5** with $[Ca]_{\text{NMDA}}$, $\eta_{s,\text{ltd}}$, $\beta_{\text{mp}}$
  - `else if dopamine_level == 0 then`
  - `# No plasticity under basal dopamine`
    - `*` $\Delta w = 0$
  - `return` $\Delta w$
- `end`

---

## Inhibitory synaptic plasticity

In contrast to the well-studied mechanistic underpinnings of glutamatergic plasticity in, e.g., SPNs, much less is known about how inhibitory synapses might be updated during learning. The inhibitory plasticity rule developed here is therefore more phenomenological and exploratory in nature and was developed to enhance nonlinearities in the local dendrite. The rule is based on the BCM formalism (*Bienenstock et al., 1982*). In the BCM rule, there is a threshold level of synaptic activity below which LTD is triggered and above which LTP is triggered (*Figure 7D*). Inspired by *Gandolfi et al., 2020*; *Chapman et al., 2022*; *Ravasenga et al., 2022*, we use calcium from all voltage-gated calcium channels ($[Ca]_v$) as the indicator of excitatory synaptic activity near the dendritic shaft where an inhibitory synapse is located. The inhibitory rule is designed to passively observe and respond to the surrounding excitatory synaptic activity, governed by local voltage-gated calcium influx, but without

**Table 2.** Inhibitory plasticity parameters.

| Parameter | Description | Value |
|---|---|---|
| $\beta_{\text{inh}}$ | Controls steepness of the curves | $2.5 \times 10^3$ mM$^{-1}$ |
| $\eta_{\text{act}}$ | Learning rate for active/inactive inhibitory weight | active: $-0.055$ , inactive: $0.055$ µS$^{-1}$ ms$^{-1}$ |
| $a_{\text{inh}}$ | Controls inhibitory weight | $-1$ |
| $b_{\text{inh}}$ | Controls inhibitory weight | $3$ |
| $w_{\text{inh}}^{\max}$ | Maximum synaptic strength | $0.005$ µS |
| $\eta_{\text{inh,high}}$ | Learning rate for modifying $\theta_{\text{inh,high}}$ | $9 \times 10^{-4}$ mM ms$^{-1}$ |
| $a_{\text{inh,high}}$ | Controls $\theta_{\text{inh,high}}$ | $-1$ |
| $b_{\text{inh,high}}$ | Controls $\theta_{\text{inh,high}}$ | $3$ |
| $\eta_{\text{inh,low}}$ | Learning rate for modifying $\theta_{\text{inh,low}}$ | $-5 \times 10^{-5}$ mM ms$^{-1}$ |
| $a_{\text{inh,low}}$ | Controls $\theta_{\text{inh,low}}$ | $-2$ |
| $b_{\text{inh,low}}$ | Controls $\theta_{\text{inh,low}}$ | $3$ |
| $c$ | Calcium concentration offset | $6 \times 10^{-4}$ mM |

reliance on dopamine or explicit feedback. It operates at a slower pace to ensure that it follows the average excitatory activity levels and is meant to enhance the contrast in activity by amplifying local differences in excitatory synaptic efficacy.

*Equation 6* describes how the inhibitory synaptic weights change based on the calcium concentration (calculated from calcium influx from all voltage-gated ion channels close to the synapse).

$$\Omega(Ca, \theta_{\text{inh,low}}, \theta_{\text{inh,high}}, a, b) = \frac{a}{1 + e^{-\beta_{\text{inh}}(Ca - \theta_{\text{inh,low}})}} + \frac{b}{1 + e^{-\beta_{\text{inh}}(Ca - \theta_{\text{inh,high}})}} \qquad (6)$$

It consists of two sigmoidal terms, where the parameters *a* and *b* determine the contribution of each component, $\beta_{\text{inh}}$ determines the steepness of the transitions, and $\theta_{\text{inh,low}}$ and $\theta_{\text{inh,high}}$ determine the calcium concentration at which each sigmoid is half activated (see *Figure 7D*). Based on this, and modulated by a rate constant, $\eta_{\text{act}}$ (*Equation 7*) describes the rule used to update the weights of the inhibitory synapses.

$$\Delta w_{\text{inh}} = \eta_{\text{act}} \cdot \Omega(Ca_v, \theta_{\text{inh,low}}, \theta_{\text{inh,high}}, a_{\text{inh}}, b_{\text{inh}}) \cdot w_{\text{inh}} \cdot (w_{\text{inh}}^{\text{max}} - w_{\text{inh}}) \qquad (7)$$

The weight change also depends on the magnitude of the weights themselves, in such a way that intermediate weights update faster than small or large weights. This helps stabilize the weights of the inhibitory synapses at the end of learning and prevents the weight from taking on large or negative values. Large weights are then close to $w_{\text{inh}}^{\text{max}}$ at the end of learning and small weights close to 0 (see *Figure 5D* for an example and *Table 2* for parameter values).

The learning rule further differentiates between active and inactive synapses. For active synapses: (i) if the calcium concentration exceeds $\theta_{\text{inh,high}}$, the weight is depressed, (ii) if calcium falls between $\theta_{\text{inh,low}}$ and $\theta_{\text{inh,high}}$, the weight is potentiated, and (iii) when calcium is below $\theta_{\text{inh,low}}$, little or no weight change occurs. For inactive synapses, the rule is reversed: (i) when calcium exceeds $\theta_{\text{inh,high}}$, the weight is potentiated, (ii) if calcium falls between $\theta_{\text{inh,low}}$ and $\theta_{\text{inh,high}}$, the weight is depressed, and (iii) no changes occur when calcium is below $\theta_{\text{inh,low}}$ (as in the case of active synapses). The schematic in *Figure 7D* (top panel) illustrates these weight updates, showing how the learning rule differentially affects active and inactive synapses. The black curves represent the relationship between calcium concentration and synaptic weight change, with the beige and blue shaded regions indicating potentiation and depression zones for each case.

The half-activation thresholds, $\theta_{\text{inh,low}}$ and $\theta_{\text{inh,high}}$, are further updated to track the calcium levels of the synapse. These updates of the thresholds ensure that the plasticity rule remains sensitive to changes in calcium concentration and stabilizes synaptic learning. *Equation 8* and *Equation 9* describe these adaptive dynamics. This is, in fact, metaplasticity of the inhibitory synapses.

$$\Delta\theta_{\text{inh,high}} = \eta_{\text{inh,high}} \cdot \Omega(Ca_v, \theta_{\text{inh,low}}, \theta_{\text{inh,high}}, a_{\text{inh,high}}, b_{\text{inh,high}}) \qquad (8)$$

$$\Delta\theta_{\text{inh,low}} = \eta_{\text{inh,low}} \cdot \Omega(Ca_v + c, \theta_{\text{inh,low}}, \theta_{\text{inh,high}}, a_{\text{inh,low}}, b_{\text{inh,low}}) \qquad (9)$$

The upper threshold $\theta_{\text{inh,high}}$ shifts toward the highest observed calcium level in the synapse, while the lower threshold $\theta_{\text{inh,low}}$ moves to a level below the maximum. The rate of these adjustments is determined by the parameters $\eta_{\text{inh,high}}$ and $\eta_{\text{inh,low}}$, which scale the influence of calcium activity on the threshold dynamics. The parameters $a_{\text{inh,high}}, b_{\text{inh,high}}$ determine the rate of change of $\theta_{\text{inh,high}}$, while $a_{\text{inh,low}}, b_{\text{inh,low}}$ define the rate of change of $\theta_{\text{inh,low}}$ (see *Table 2* for parameter values). The parameter *c* in *Equation 9* is used to make the two curves for the high and low thresholds ($\theta_{\text{inh,low}}, C_{\text{inh,high}}$) intersect at zero which minimizes the fluctuations around the transition point (green asterisk in *Figure 7D*, bottom panel).

## Pseudocode for the inhibitory plasticity rule

**Inputs:**

- $[Ca]_V$: Ca concentration from voltage-gated calcium channels

**Output:**

- `return` $\Delta w_{\text{inh}}$: Change in synaptic weight

*Continued on next page*

---

Algorithm 2:

- begin
  - if synapse_active then
  - # Active synapse:  Compute weight change using negative learning rate $(\eta_{act})$
    * $\Delta w_{inh}$ = solve *Equation 7* with $[Ca]_V$, $\eta_{act}$, $\theta_{inh,high}$, $\theta_{inh,low}$, $a_{inh}$, $b_{inh}$, $W_{inh}^{max}$
    * update $\theta_{inh,high}$ using *Equation 8* with $\theta_{inh,high}$, $\theta_{inh,low}$, $\eta_{inh,high}$, $a_{inh,high}$, $b_{inh,high}$
    * update $\theta_{inh,low}$ using *Equation 9* with $\theta_{inh,high}$, $\theta_{inh,low}$, $\eta_{inh,low}$, $a_{inh,low}$, $b_{inh,low}$, c
  - else
  - # Inactive synapse:  Compute weight change using positive learning rate $(\eta_{act})$
    * $\Delta w_{inh}$ = solve *Equation 7* with $[Ca]_V$, $\eta_{act}$, $\theta_{inh,high}$, $\theta_{inh,low}$, $a_{inh}$, $b_{inh}$, $W_{inh}^{max}$
    * update $\theta_{inh,high}$ using *Equation 8* with $\theta_{inh,high}$, $\theta_{inh,low}$, $\eta_{inh,high}$, $a_{inh,high}$, $b_{inh,high}$
    * update $\theta_{inh,low}$ using *Equation 9* with $\theta_{inh,high}$, $\theta_{inh,low}$, $\eta_{inh,low}$, $a_{inh,low}$, $b_{inh,low}$, c
  - return $\Delta w_{inh}$
- end

---

## Training procedure

The features of a stimulus were represented by the shape and color of bananas and strawberries. We presented the neuron with a sequence of stimuli, typically around 960, each belonging to one of four possible feature combinations. The four stimuli: 'red strawberry', 'yellow banana', 'red banana', and 'yellow strawberry' were presented in random order three times each within a block of 12 stimuli, followed by another reshuffled block of 12, and so on. Each stimulus presentation lasted 20 ms, during which time all the stimuli-related synapses received one randomly timed spike per synapse (see *Figure 3B*). Whenever relevant, inhibitory synapses were activated simultaneously with the excitatory synapses, within a window of 100 ms. The stimuli presentation was followed by a reward cue lasting 50 ms that arrived 300 ms after the stimulus onset. The reward cue was represented with +1 for relevant stimuli if the neuron spiked (representing a peak in dopamine), with –1 for irrelevant stimuli if the neuron spiked (representing a dopamine pause) and 0 for baseline levels or when the neuron was silent. However, in an additional subthreshold learning task, the reward cue was delivered even in the absence of somatic spiking for active glutamate synapses, such that if they were active following the relevant stimuli ('red strawberry' and 'yellow banana') they got a dopamine peak and a dopamine pause for irrelevant stimuli (*Figure 6—figure supplement 1B and C*). The time between two stimuli was 800 ms, long enough to allow for the voltage, calcium, and all other state variables in the model to return to their baseline values (*Figure 3B*). During the stimulus presentation, the stimuli-related synapses receive one randomly timed spike per synapse. Further, the learning rule was 'on' during the whole procedure, such that synapses were continuously updated throughout the simulation. Hence, there were no separate training and testing phases in which synapses were plastic and frozen, respectively.

## Clustered setup

This setup was based on the assumption of pre-existing clustered synapses for each feature. Features were allocated to two dendritic branches. Each branch had each feature represented with five synapses clustered closely on a single dendrite. Depending on the feature combination, two, three, or four features were represented in clusters on one or both dendritic branches (see *Figure 3A*, *Figure 4A*, and *Figure 4D* for examples). Additionally, 108 feature-unspecific synapses were distributed throughout the dendrites, activated concurrently with all stimuli to enhance the probability of spiking (as plateau potentials together with the general background synaptic noise used do not often lead to spikes in SPNs, *Figure 2A*). The features representing a stimulus (clustered synapses) were active within 20 ms, while feature-unspecific input was activated over a 50 ms window. In simulations with inhibition, four inhibitory synapses, each one representing a feature, were placed near each cluster (*Figure 5A*). Within this setup, a single stimulus activated both excitatory and inhibitory synapses. To match the level of depolarization seen in our excitatory-only setup, the number of feature-unspecific synapses was increased to 144 in the simulations including inhibitory plasticity. The initial conductance of inhibitory synapses was set to 0.1 ± 0.01 nS.

## Distributed setup

In contrast, this setup examined learning dynamics in neurons without pre-existing synaptic clustering for individual features. A total of 200 excitatory synapses were randomly distributed over 30

dendrites. Each feature was represented by 40 excitatory synapses, and an additional 40 feature-unspecific excitatory synapses were used (*Figure 6A*). We initiated this experiment with excitatory synaptic weights at 0.3 ± 0.1 (around 0.75 nS). This higher initial weight was chosen to compensate for the reduced efficacy of non-clustered synaptic inputs in producing sufficient depolarization and calcium influx. Extending our investigation, we added 60 inhibitory synapses, 15 for each feature, dispersed randomly over the 30 dendrites. We initiated this extended experiment with excitatory synaptic weights at 0.45 ± 0.1 (around 1.125 nS), aiming to maintain the same baseline voltage activity as in the excitatory-only case. *Figure 6A* illustrates this setup, exemplifying the pre- and post-learning synaptic weights for both excitatory and inhibitory synapses.

## Acknowledgements

We would like to thank the members of the Hellgren Kotaleski laboratory for their helpful discussions on various aspects of the manuscript. DT extends gratitude to Ana Kalajdjieva for illustrating the mouse brain. We acknowledge the use of Fenix Infrastructure resources, which are partially funded by the European Union's Horizon 2020 research and innovation program through the ICEI project under grant agreement No. 800858. Simulations were also performed on resources provided by the National Academic Infrastructure for Supercomputing in Sweden (NAISS) at PDC KTH, partially funded by the Swedish Research Council through grant agreement No. 2022-06725. This study was supported by the Swedish Research Council (VR-M-2020-01652), the Swedish e-Science Research Centre (SeRC), Science for Life Laboratory, EU/Horizon 2020 No. 945539 (HBP SGA3) and No. 101147319 (EBRAINS 2.0 Project), the European Union's Research and Innovation Program Horizon Europe under grant agreement No. 101137289 (the Virtual Brain Twin Project), and KTH Digital Futures.

## Additional information

### Funding

| Funder | Grant reference number | Author |
| --- | --- | --- |
| Vetenskapsrådet | VR-M-2020-01652 | Jeanette Hellgren Kotaleski |
| Swedish e-Science Research Centre | SeRC support | Jeanette Hellgren Kotaleski |
| Science for Life Laboratory | institutional support | Jeanette Hellgren Kotaleski |
| European Commission | 10.3030/945539 | Jeanette Hellgren Kotaleski |
| European Commission | 10.3030/101147319 | Jeanette Hellgren Kotaleski |
| European Commission | 10.3030/101137289 | Jeanette Hellgren Kotaleski |
| Digital Futures | institutional support | Jeanette Hellgren Kotaleski |

The funders had no role in study design, data collection and interpretation, or the decision to submit the work for publication.

### Author contributions

Zahra Khodadadi, Conceptualization, Data curation, Software, Formal analysis, Validation, Investigation, Visualization, Methodology, Writing – original draft, Writing – review and editing; Daniel Trpevski, Data curation, Software, Formal analysis, Validation, Visualization, Methodology, Writing – original draft, Writing – review and editing, Conceptualization; Robert Lindroos, Supervision, Validation, Visualization, Writing – original draft, Writing – review and editing; Jeanette Hellgren Kotaleski, Conceptualization, Resources, Supervision, Funding acquisition, Writing – original draft, Project administration, Writing – review and editing

## Author ORCIDs
Zahra Khodadadi ⓘ https://orcid.org/0000-0001-6124-949X
Daniel Trpevski ⓘ https://orcid.org/0000-0001-9068-6744
Robert Lindroos ⓘ https://orcid.org/0000-0002-9134-3601
Jeanette Hellgren Kotaleski ⓘ https://orcid.org/0000-0002-0550-0739

Reviewer #1 (Public review): https://doi.org/10.7554/eLife.97274.4.sa1
Reviewer #2 (Public review): https://doi.org/10.7554/eLife.97274.4.sa2
Author response https://doi.org/10.7554/eLife.97274.4.sa3

---

# Additional files

### Supplementary files
MDAR checklist

### Data availability
This manuscript is a computational study. No new experimental data were generated. All code used for simulations, analyses, and figure generation (including figure supplements) is publicly available at https://github.com/zahradd/dSPN-learning-rule (copy archived at *Khodadadi, 2025*). This repository includes the simulations underlying the figures in the manuscript, along with a README file providing detailed instructions for environment setup and result reproduction. Additionally, model components adapted from *Lindroos and Hellgren Kotaleski, 2021* and *Trpevski et al., 2023*, are available on ModelDB (accession numbers: 266775 and 2017143).

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
