## [Editor Report · eLife Assessment]

This computational modeling study builds on multiple previous lines of experimental and theoretical research to investigate how a single neuron can solve a nonlinear pattern classification task. The revised manuscript presents **convincing** evidence that the location of synapses on dendritic branches, as well as synaptic plasticity of excitatory and inhibitory synapses, influences the ability of a neuron to discriminate combinations of sensory stimuli. The ideas in this work are very interesting, presenting an **important** direction in the computational neuroscience field about how to harness the computational power of "active dendrites" for solving learning tasks.

---

## [Referee Report · Reviewer #1 (Public review)]

Summary:

This computational modeling study builds on multiple previous lines of experimental and theoretical research to investigate how a single neuron can solve a nonlinear pattern classification task. The authors construct a detailed biophysical and morphological model of a single striatal medium spiny neuron, and endow excitatory and inhibitory synapses with dynamic synaptic plasticity mechanisms that are sensitive to (1) the presence or absence of a dopamine reward signal, and (2) spatiotemporal coincidence of synaptic activity in single dendritic branches. The latter coincidence is detected by voltage-dependent NMDA-type glutamate receptors, which can generate a type of dendritic spike referred to as a "plateau potential." In the absence of inhibitory plasticity, the proposed mechanisms result in good performance on a nonlinear classification task when specific input features are segregated and clustered onto individual branches, but reduced performance when input features are randomly distributed across branches. Interestingly, adding inhibitory plasticity improves classification performance even when input features are randomly distributed.

Strengths:

The integrative aspect of this study is its major strength. It is challenging to relate low-level details such as electrical spine compartmentalization, extrasynaptic neurotransmitter concentrations, dendritic nonlinearities, spatial clustering of correlated inputs, and plasticity of excitatory and inhibitory synapses to high-level computations such as nonlinear feature classification. Due to high simulation costs, it is rare to see highly biophysical and morphological models used for learning studies that require repeated stimulus presentations over the course of a training procedure. The study aspires to prove the principle that experimentally-supported biological mechanisms can explain complex learning.

Weaknesses:

The high level of complexity of each component of the model makes it difficult to gain an intuition for which aspects of the model are essential for its performance, or responsible for its poor performance under certain conditions. Stripping down some of the biophysical detail and comparing it to a simpler model may help better understand each component in isolation.

---

## [Referee Report · Reviewer #2 (Public review)]

Summary:

The study explores how single striatal projection neurons (SPNs) utilize dendritic nonlinearities to solve complex integration tasks. It introduces a calcium-based synaptic learning rule that incorporates local calcium dynamics and dopaminergic signals, along with metaplasticity to ensure stability for synaptic weights. Results show SPNs can solve the nonlinear feature binding problem and enhance computational efficiency through inhibitory plasticity in dendrites, emphasizing the significant computational potential of individual neurons. In summary, the study provides a more biologically plausible solution to single-neuron learning and gives further mechanical insights into complex computations at the single-neuron level.

Strengths:

The paper introduces a novel learning rule for training a single multicompartmental neuron model to perform nonlinear feature binding tasks (NFBP), highlighting two main strengths: the learning rule is local, calcium-based, and requires only sparse reward signals, making it highly biologically plausible, and it applies to detailed neuron models that effectively preserve dendritic nonlinearities, contrasting with many previous studies that use simplified models.

---

## [Author Response]

The following is the authors’ response to the previous reviews

**Reviewer #1 (Public review):**
Summary:This computational modeling study builds on multiple previous lines of experimental and theoretical research to investigate how a single neuron can solve a nonlinear pattern classification task. The authors construct a detailed biophysical and morphological model of a single striatal medium spiny neuron, and endow excitatory and inhibitory synapses with dynamic synaptic plasticity mechanisms that are sensitive to (1) the presence or absence of a dopamine reward signal, and (2) spatiotemporal coincidence of synaptic activity in single dendritic branches. The latter coincidence is detected by voltage-dependent NMDA-type glutamate receptors, which can generate a type of dendritic spike referred to as a "plateau potential." In the absence of inhibitory plasticity, the proposed mechanisms result in good performance on a nonlinear classification task when specific input features are segregated and clustered onto individual branches, but reduced performance when input features are randomly distributed across branches. Interestingly, adding inhibitory plasticity improves classification performance even when input features are randomly distributed.Strengths:The integrative aspect of this study is its major strength. It is challenging to relate low-level details such as electrical spine compartmentalization, extrasynaptic neurotransmitter concentrations, dendritic nonlinearities, spatial clustering of correlated inputs, and plasticity of excitatory and inhibitory synapses to high-level computations such as nonlinear feature classification. Due to high simulation costs, it is rare to see highly biophysical and morphological models used for learning studies that require repeated stimulus presentations over the course of a training procedure. The study aspires to prove the principle that experimentally-supported biological mechanisms can explain complex learning.Weaknesses:The high level of complexity of each component of the model makes it difficult to gain an intuition for which aspects of the model are essential for its performance, or responsible for its poor performance under certain conditions. Stripping down some of the biophysical detail and comparing it to a simpler model may help better understand each component in isolation.

We greatly appreciate your recognition of the study’s integrative scope and the challenges of linking detailed biophysics to high-level computation. We acknowledge that the model’s complexity can obscure the contribution of individual components. However, as stated in the introduction the principles already have been shown in simplified theoretical models for instance in Tran-Van-Minh et al. 2015. Our aim here was to extend those ideas into a more biologically detailed setting to test whether the same principles still hold under realistic constraints. While simplification can aid intuition, we believe that demonstrating these effects in a biophysically grounded model strengthens the overall conclusion. We agree that further comparisons with reduced models would be valuable for isolating the contribution of specific components and plan to explore that in future work.

**Reviewer #2 (Public review):**
Summary:The study explores how single striatal projection neurons (SPNs) utilize dendritic nonlinearities to solve complex integration tasks. It introduces a calcium-based synaptic learning rule that incorporates local calcium dynamics and dopaminergic signals, along with metaplasticity to ensure stability for synaptic weights. Results show SPNs can solve the nonlinear feature binding problem and enhance computational efficiency through inhibitory plasticity in dendrites, emphasizing the significant computational potential of individual neurons. In summary, the study provides a more biologically plausible solution to single-neuron learning and gives further mechanical insights into complex computations at the single-neuron level.Strengths:The paper introduces a novel learning rule for training a single multicompartmental neuron model to perform nonlinear feature binding tasks (NFBP), highlighting two main strengths: the learning rule is local, calcium-based, and requires only sparse reward signals, making it highly biologically plausible, and it applies to detailed neuron models that effectively preserve dendritic nonlinearities, contrasting with many previous studies that use simplified models.

Thank you for highlighting the biological plausibility of our calcium- and dopamine-dependent learning rule and its ability to exploit dendritic nonlinearities. Your positive assessment reinforces our commitment to refining the rule and exploring its implications in larger, more diverse settings.

**Reviewer #1 (Recommendations for the authors):**
Major recommendations:P9: When introducing the excitatory learning rule, the reader is referred to the Methods. I suggest moving Figure 7A-D, "Excitatory plasticity" to be more prominently presented in the main body of the paper where the reader needs to understand it. There are errors in the current Figure 7, and wrong/confusing acronyms. The abbreviations "LTP-K" and "MP-K" are not intuitive. In A, I would spell out "LTP kernel" and "Theta_LTP adaptation". In B, I would spell out "LTD kernel" and "Theta_LTD adaptation".

We have clarified the terminology in Figure 7 by replacing “LTP-K” with “LTP kernel” and “MP-K” with “metaplasticity kernel”. While we kept Figure 7 in the Methods section to maintain the flow of the main text, we agree that an earlier introduction of the learning rule improves clarity. To that end, we added a simplified schematic to Figure 3 in the Results section, which provides readers with an accessible overview of the excitatory plasticity mechanism at the point where it is first introduced.

In C, for simplicity and clarity, I would only show the initial and updated LTP kernel and Calcium and remove the Theta_LTP adaptation curve, it's too busy and not necessary. Similarly in D, I would show only the initial and updated LTD kernel and Calcium and remove the Theta_LTD adaptation curve. In the current version of the Figure, panel B, right incorrectly labels "Theta_LTD" as "Theta_LTP". Panel D incorrectly labels "LTD kernel" as "LTP/MP-K" in the subheading and "MP/LTP-K" in the graph.

To avoid confusion and better illustrate the interactions between calcium signals, kernels, and thresholds, we have added a movie showing how these components evolve during learning. The figure panels remain as originally designed, since the LTP kernel governs both potentiation and depression through metaplastic threshold adaptation, while the LTD kernel remains fixed.

P17: Again, instead of pointing the reader to the Methods, I would move Figure 7E, "Inhibitory plasticity" to the main body of the paper where the reader needs to understand it. For clarity, I would label "C_TL" and "Theta_Inh,low" and "C_TH" as "Theta_Inh,high". The right panel could be better labeled "Inhibitory plasticity kernel". The left panel could be better labeled "Theta_Inh adaptation", with again replacing the acronyms "C_TL" and "C_TH". The same applies to Fig. 5D on P19.

We have updated the labeling in Figures 5D and 7E for clarity, including replacing "C_TL" and "C_TH" with "Theta_Inh,low" and "Theta_Inh,high". In addition, we added a simplified schematic of the inhibitory plasticity rule to Figure 5 to assist the reader’s understanding when presenting the results. Figure 7E remains in the Methods section to preserve the flow of the main text.

P12: I would suggest simplifying Fig. 3 panels and acronyms as well. Remove "MP-K" from C and D. Relabel "LTP-K" as "LTP kernel". The same applies to Fig. 5E on P19 and Fig. 3 - supplement 1 on P46 and Fig 6 - supplement 1 on P49.

We have simplified the labeling across all relevant figures by replacing “MP-K” with “metaplasticity kernel” and “LTP-K” with “LTP kernel.” To maintain clarity, we retained these terms in only one panel as a reference.

Minor recommendations:P4: "Although not discussed much in more theoretical work, our study demonstrates the necessity of metaplasticity for achieving stable and physiologically realistic synaptic weights." This sentence is jarring. BCM and metaplasticity has been discussed in hundreds of theory papers! Cite some. This sentence would more accurately read, "Our study corroborates prior theory work (citations) demonstrating that metaplasticity helps to achieve stable and physiologically realistic synaptic weights."

We have followed the reviewers suggestion and updated the sentence to: Previous theoretical studies (Bienenstock et al., 1982; Fusi et al., 2005; Clopath et al., 2010; Benna & Fusi, 2016; Zenke & Gerstner, 2017) demonstrate the essential role of metaplasticity in maintaining stability in synaptic weight distributions. (page 2 line 49-51, page 3 line 1)

P9: Grammar. "The neuron model was during training activated..." should read "During training, the neuron model was activated..."

Corrected

P17: Lovett-Barron et al., 2012 is appropriately cited here. Milstein et al., Neuron, 2015 also showed dendritic inhibition regulates plateau potentials in CA1 pyramidal cells in vitro, and Grienberger et al., Nat. Neurosci., 2017 showed it in vivo.P19 vs P16 vs P21. Fig. 4B, Fig. 5B, and Fig. 6B choose different strategies to show variance across seeds. Please choose one strategy and apply to all comparable plots.

We thank the reviewer for these helpful points.

We have added the suggested citations (Milstein et al., 2015; Grienberger et al., 2017) alongside Lovett-Barron et al., 2012.

Variance across seeds is now displayed uniformly (mean is solid line STD is shaded area) in Figures 4B, 5B, and 6B.

**Reviewer #2 (Recommendations for the authors):**
Major Points:(1) Quality of Scientific Writing:i. Mathematical and Implementation Details:I appreciate the authors' efforts in clarifying the mathematical details and providing pseudocode for the learning rule, significantly improving readability and reproducibility. The reference to existing models via GitHub and ModelDB repositories is acceptable. However, I suggest enhancing the presentation quality of equations within the Methods section-currently, they are low-resolution images. Please consider rewriting these equations using LaTeX or replacing them with high-resolution images to further improve clarity.

We appreciate the reviewer’s comment regarding clarity and reproducibility. In response, we have rewritten all equations in LaTeX to improve their readability and presentation quality in the Methods section.

ii. Figure quality.I acknowledge the authors' effort to improve figure clarity and consistency throughout the manuscript. However, I notice that the x-axis label "[Ca]_v (μm)" in Fig. 7E still appears compressed and unclear. Additionally, given the complexity and abundance of hyperparameters or artificial settings involved in your experimental design and learning rule (such as kernel parameters, metaplasticity kernels, and unspecific features), the current arrangement of subfigures (particularly Fig. 3C, D and Fig. 5D, E) still poses readability challenges. I recommend reordering subfigures to present primary results (e.g., performance outcomes) prominently upfront, while relegating visualizations of detailed hyperparameter manipulations or feature weight variations to later sections or the discussion, thus enhancing clarity for readers.

We thank the reviewer for pointing out the readability issue. We have corrected the x-axis label in Figure 7D. We hope this new layout with a simplified rule in Fig 3 and Fig 5 presents the key findings while retaining full mechanistic detail to make it easier to understand the model behavior.

iii. Writing clarity.The authors have streamlined the "Metaplasticity" section and reduced references to dopamine, which is a positive step. However, the broader issue remains: the manuscript still appears overly detailed and more like a technical report of a novel learning rule, rather than a clearly structured scientific paper. I strongly recommend that the authors further distill the manuscript by clearly focusing on one or two central scientific questions or hypotheses-for instance, emphasizing core insights such as "inhibitory inputs facilitate nonlinear dendritic computations" or "distal dendritic inputs significantly contribute to nonlinear integration." Clarifying and highlighting these primary scientific questions early and consistently throughout the manuscript would substantially enhance readability and impact.

We appreciate the reviewer’s guidance on improving the manuscript’s clarity and focus.In response, we now highlight two central questions at the end of the Introduction and have retitled the main Results subsections to follow this thread, thereby sharpening the manuscript’s focus while retaining necessary technical detail (page3 line 20-28).We have also removed redundant passages and simplified technical details to improve overall readability .

Minor:(1) The [Ca]NMDA in Figure 2A and 2C can have large values even when very few synapses are activated. Why is that? Is this setting biologically realistic?The authors acknowledge that their simulated [Ca²⁺] levels exceed typical biological measurements but claim that the learning rule remains robust across variations in calcium concentrations. However, robustness to calcium variations was not explicitly demonstrated in the main figures. To convincingly address this concern, I recommend the authors explicitly test and present whether adopting biologically realistic calcium concentrations (~1 μM) impacts the learning outcomes or synaptic weight dynamics. Clarifying this point with a supplemental analysis or an additional figure panel would significantly strengthen their argument regarding the model's biological plausibility and robustness.

We thank the reviewer for the comment. The elevated [Ca^²⁺^]_NMDA_ values reflect localized transients in spine heads with narrow necks and high NMDA conductance. These values are not problematic for our model, as the plasticity rule depends on relative calcium differences rather than absolute levels as the metaplasticity kernel will adjust. In future versions of our detailed neuron model, we will likely decrease the spine axial resistance of the spine neck.